# Native globular ferritin nanopore sensor

Yun-Dong Yin[1,2,3], Yu-Wei Zhang[1,3], Xi-Tong Song[1,3], Jun Hu[1], Yu-Heng Chen[1], Wen-Chuan Lai[1], Ya-Fei Li [1] & Zhi-Yuan Gu [1] ✉

High-resolution nanopore analysis technology relies on the design of novel transmembrane protein platforms. Traditional barrel-shaped protein channels are preferred for constructing nanopore sensors, which may miss protein candidates in non-barrel structures. Here, we demonstrate the globular ferritin displays excellent membrane-insertion capacity and stable transmembrane ionic current owing to its hydrophobic four-fold channels and hydrophilic three-fold channels. The ionic current rectification and voltage-gating characteristics are discovered in single-ferritin ionic current measurement. Notably, the ferritin is used as a nanopore sensor, by which we achieve the high resolution discrimination of L-cysteine, L-homocysteine, and cysteine-containing dipeptides with the assistance of equivalent $Cu^{2+}$. The mechanistic studies by multiple controlled experiments and quantum mechanics/all-atom/ coarse-grained multiscale MD simulations reveal that analytes are synergistically captured by His114, Cys126, and Glu130 within C3 channel, causing the current blockage signals. The promising ferritin nanopore sensor provides a guide to discovering new protein nanopores without shape restrictions.

Nanopore sensing based on transmembrane proteins has exhibited great potential in the field of label-free and highly sensitive single-molecule detection[1–5]. Conventional protein nanopores such as α-HL, MspA, and aerolysin have made great progress in DNA and protein sequencing[6–8], amino acid recognition[9,10], biomarker detection[11–15], and single-molecule interactions study[16–19]. The development of new biological pores is considered a crucial means to enhance the resolution and expand the detection range of analytes. The membrane-insertion capacity (surface hydrophobicity) and ionic transport property (inside hydrophilicity) are two important indicators to evaluate whether a channel structure is suitable for constructing a nanopore sensor. Under this principle, some novel barrel-shaped nanopores including α-helical peptide pores[20–22], β-barrel nanopores[23,24], and DNA-based nanopores[25–27] were designed and constructed. Usually, to enhance the hydrophobicity, the de novo pore design necessitates complex chemistry involving unnatural residues or tags incorporation. In a traditional perspective, the barrel structured channel with hydrophobic surface and hydrophilic inwall[28,29] is preferred in the development of novel nanopores. However, natural proteins with non-barrel shape but possess promising membrane-insertion capacity and ionic transport properties remain largely unexplored.

Ferritin is a globular protein composed of 24 subunits with the function of storing and releasing iron to maintain the iron balance in living organisms[30–33]. To our best knowledge, the ferritin, with large size and globular configuration, was never reported to be inserted into the lipid bilayer and function as a transmembrane channel, let alone for single-molecule sensing. However, viewing from membrane-insertion capacity, the hydrophobic sites including six four-fold (C4) channels on the surface of ferritin could provide the attachment points for phospholipid molecules. On the other hand, ferritin has eight hydrophilic three-fold (C3) channels with narrow aperture (3-5 Å), which are considered as the natural ion channels for iron transport[34,35], also providing the confined space for potential single-molecule sensing. Besides, there are three natural cysteine residues inside C3 channel[34], which provide the possible reactive sites for single-molecule interaction, avoiding the complicated protein mutation during nanopore engineering. Thereby, we consider ferritin as an excellent candidate for a transmembrane protein to build a novel high-resolution nanopore sensor.

[1]State Key Laboratory of Microbial Technology, Jiangsu Collaborative Innovation Center of Biomedical Functional Materials, Jiangsu Key Laboratory of New Power Batteries, College of Chemistry and Materials Science, Nanjing Normal University, Nanjing 210023, China. [2]College of Life Sciences, Nanjing Normal University, Nanjing 210023, China. [3]These authors contributed equally: Yun-Dong Yin, Yu-Wei Zhang, Xi-Tong Song. ✉e-mail: guzhiyuan@njnu.edu.cn

Here, we successfully insert single horse spleen ferritin into the planar lipid bilayer, constructing the ferritin-based transmembrane nanopore. The multiple insertion experiments and molecular dynamics (MD) simulation are used to confirm the establishment of a single-ferritin transmembrane system and evaluate the stability of the ferritin-lipid bilayer system, respectively, which indicates the ferritin is able to be individually and stably embedded in the lipid bilayer. The asymmetric *I-V* curves and voltage-dependent channel opening and closing behavior are discovered, revealing the ionic current rectification and voltage-gating characteristics of the ferritin nanopore. Furthermore, the ferritin nanopore is developed as a single-molecule sensor for amino acid and peptide detection with the assistance of $Cu^{2+}$. The detection mechanism is also explored by combining controlled nanopore experiments and multiscale MD simulations. The $Cu^{2+}$ is demonstrated to bridge analytes and residues (Cys126 and Glu130) inside the ferritin C3 channel, leading to the blockage current signals. The globular ferritin nanopore confirms that a high-resolution nanopore sensor could be constructed by non-barrel protein which greatly expands the protein candidates as nanopore sensors.

## Results and discussion
### Electrical properties and MD simulation of ferritin nanopore on the lipid bilayer

The horse spleen ferritin is assembled by 24 identical subunits, resulting in a polyhedron-like globular structure with a size of ~10 nm, as demonstrated in the previous works[32,36,37]. There are eight hydrophilic C3 channels on the vertex of ferritin and six hydrophobic C4 channels on the face center of ferritin (Fig. 1a, b). Specifically, the C3 channel is comprised of three subunits, in which three-fold Cys126, Asp127, and Glu130 residues enable a pore with the size of 3-5 Å (Fig. 1a). The C3 channel is considered as the specific channel through which ferritin absorbs and releases $Fe^{2+}$. In detail, the Cys126 residue near the external entrance of C3 channel is reported to play a key role in absorbing $Fe^{2+}$ [34]. The C3 ionic channel makes it possible to construct a ferritin nanopore with stable ionic current. To be noted that the ferritin sample used in nanopore experiments (Sigma-Aldrich) is comprehensively characterized by SEC, SDS-PAGE, Native-PAGE, DLS, and HRTEM experiments and identified as intact L-rich ferritin with spherical structure (Supplementary Fig. 1). For convenience, the

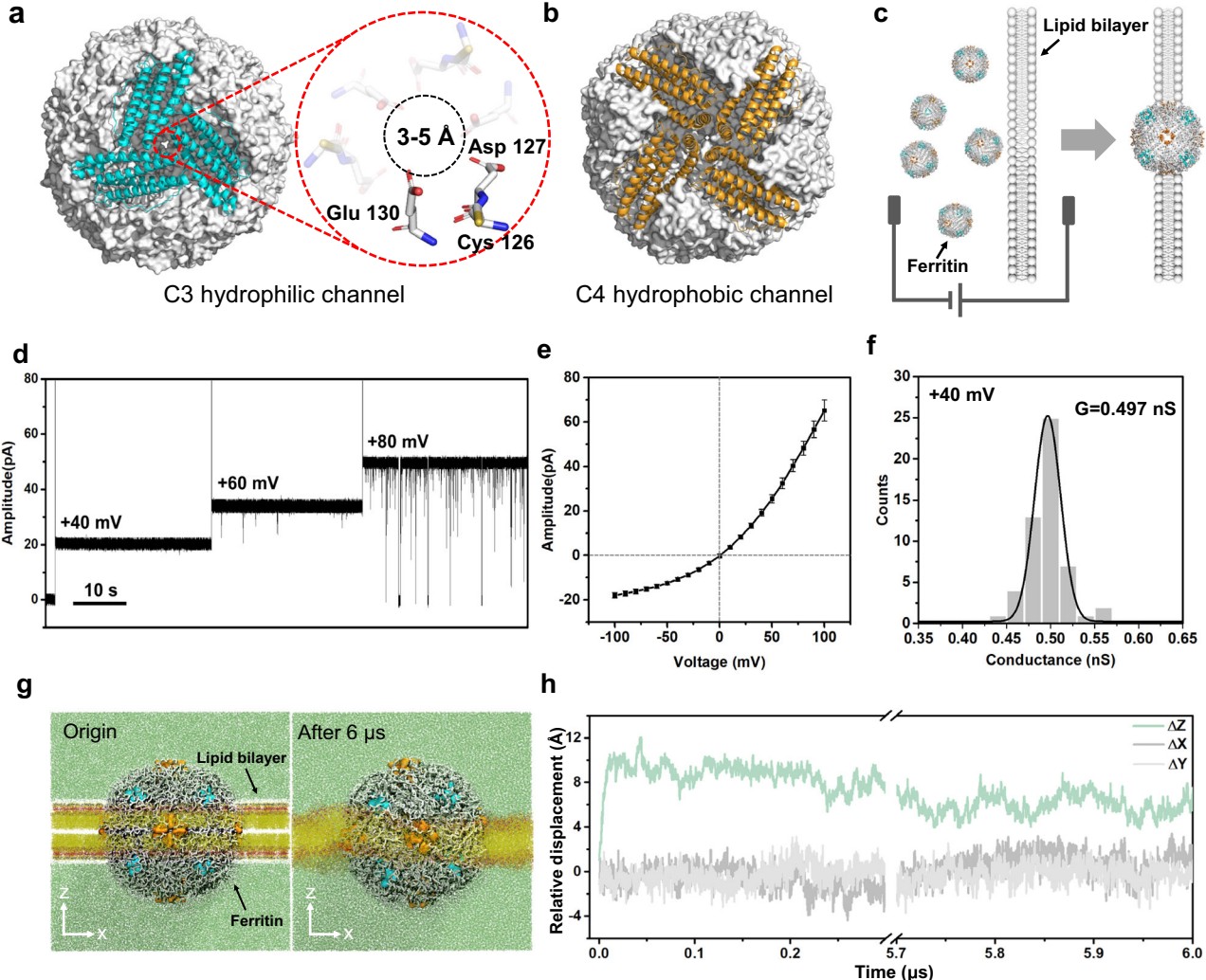

**Fig. 1 | Structure, ionic transport properties, and MD simulation of ferritin nanopore. a, b** Hydrophilic C3 channel (blue) and hydrophobic C4 channel (orange) of ferritin (grey). The C3 channel with the size of 3–5 Å is surrounded by three-folds of Cys126, Asp127, and Glu130, which plays the key role in ion transport. **c** Cartoon models of inserting ferritin nanopore into the lipid bilayer. **d** Electrical recording of ferritin nanopore showing voltage-dependent channel opening and closing. **e** *I − V* curve obtained from a single ferritin nanopore with nine independent experiments. The centre and error bar represent mean value and standard deviation (SD), respectively. **f** Histogram and Gaussian fitting of channel conductance obtained from 53 independent single-ferritin recordings at +40 mV. **g** The multiscale model of ferritin-lipid bilayer system before (left) and after (right) MD equilibrium. The fatty tails and the head group of 1,2-diphytanoyl-sn-glycero-3-phosphocholine (DPhPC) are colored yellow and red, respectively. Water molecules are marked green. **h** Relative displacement along three directions between the mass center of ferritin and all phospholipid molecules.

"ferritin" was used to represent the L-rich horse spleen ferritin in this work.

The ferritin nanopore sensing apparatus is prepared by inserting one ferritin molecule into the planar lipid bilayer, where four C4 channels are embedded in the lipid bilayer, and eight C3 channels are distributed on both sides of the membrane, with four on each side. (Fig. 1c and Supplementary Fig. 2). The abrupt current increasing shown in Supplementary Fig. 3 indicates the successful construction of ferritin nanopore. The consistent pore-forming currents in three independent nanopore insertion experiments demonstrate the reliability of the ferritin nanopore (Supplementary Fig. 4). After that, the electrical properties of the ferritin nanopore are characterized. The voltage-gated characteristic of ferritin nanopore is discovered, which maintains the open state at a low voltage of +40 mV but starts to randomly close at a high voltage of +80 mV (Fig. 1d). To be noted that, all the electrophysiological recordings in this work, if without special illustration, are carried out at +40 mV to avoid the random opening-closing background signals, despite the background signals at higher voltage of +80 mV have insignificant interference in detecting analytes (Supplementary Fig. 16). In addition, the asymmetric I-V curve of ferritin nanopore is observed, exhibiting larger conductance at positive voltages and smaller conductance at negative voltages (Fig. 1e). Several insertion experiments exhibit the same I-V curves with a specific current of -65 pA at +100 mV and ~-18 pA at -100 mV, revealing the uniformity of ferritin nanopore. (Fig. 1e and Supplementary Fig. 5). The further insertion of a second ferritin causes a double ionic current and keeps the specific I-V characteristic of ferritin nanopore (Supplementary Fig. 6), revealing that the special I-V curve shown as Fig. 1e means only one ferritin was inserted into the membrane. The ferritin nanopore has a mean conductance (G) of $0.497 \pm 0.045$ nS at +40 mV, based on the statistical analysis of $n = 53$ independent experiments (Fig. 1f). This conductance corresponds to ~$1.25 \times 10^8$ ions per second, which is consistent with the magnitude of ion channels[38]. The ionic current is believed to result from ions passing through only C3 channels rather than hydrophobic C4 channels, according to the reported ions transport pathway of ferritin[34]. The above asymmetric I-V curve, voltage-gating behavior, and high ion flux are the features of ion channels[38–40], confirming the successful construction of a single ferritin nanopore on the lipid bilayer. It is worth noting that the L-subunit of ferritin appears to play a key role in facilitating the embedding of ferritin into lipid bilayer. In our experiments, both L-rich horse ferritin and recombinantly expressed human L-ferritin (LFn) could successfully embed into lipid bilayer, whereas recombinantly expressed human H-ferritin (HFn) consistently failed to incorporate despite multiple attempts.

To evaluate the stability of the ferritin-lipid bilayer system, the coarse-grained molecular dynamics (CG-MD) simulation[41–46] is employed. (Please refer to Methods for modeling details). The initial conformation is shown on the left side of Fig. 1g. The ferritin is placed at the center of the 1,2-diphytanoyl-sn-glycero-3-phosphocholine (DPhPC) bilayer, and the four C4 hydrophobic channels are located on the XY plane in the middle of the lipid bilayer. The system is then relaxed by the isobaric-isothermal ensemble (NPT ensemble)[47,48] for 6.0 μs, with the reference temperature set to 373 K to speed up conformational changes. For the first 0.05 μs, the embedding ferritin exhibits obvious movement relative to the lipid bilayer on the Z-direction. As shown in Fig. 1h, the mass center of ferritin moves away from that of the bilayer along the Z-direction and quickly reaches equilibrium, meanwhile having almost no displacement along the X and Y axes, indicating that the ferritin can be stably embedded in the lipid bilayer. The final conformation is shown on the right side of Fig. 1g. In the simulation system, four of six hydrophobic C4 channels are always embedded in the phospholipid membrane and all eight hydrophilic C3 channels are exposed on both sides of the bilayer, directly contacting with the solution. The C3 channels exposed in the solution are considered to be the channels leading the ionic current.

## Single-molecule sensing with ferritin nanopore

The dimensions of most common protein nanopores are larger than 7 Å[49], while the C3 channel of ferritin has a smaller channel size (3-5 Å), which could render a higher spatial resolution. In addition, the cysteine mutation is most used in nanopore engineering to enhance the reactivity of nanopores[50–52], while there are three native cysteines in the C3 channel, enabling ferritin nanopore a good reactivity even without mutation. Therefore, the ferritin nanopore can be developed as a single-molecule sensor with high resolution and reactivity. Both L-Cysteine (L-Cys) and L-Homocysteine (L-Hcy) are significant biothiols that are considered as biomarkers related to colorectal cancer and cardiovascular diseases[53,54]. However, the slight difference (only one methylene) between L-Cys and L-Hcy causes similar physical and chemical properties, leading to a difficult discrimination between them. Lu et al. successfully differentiated L-Cys and L-Hcy assisted by synthetic complex probe with mutant AeL nanopore[55]. Cao et al. also achieved the differentiation between them by introducing $[AuCl_4]^-$ as a bridge with methionine-mutated MspA nanopore, in which the sulfur atom on methionine plays a key role[56]. With three reactive cysteine residues in each C3 channel, the ferritin nanopore is considered as a proper sensor for the discrimination between L-Cys and L-Hcy.

Based on as-constructed ferritin nanopore sensor, direct measurement of L-Cys (80 μM) in the KCl buffer (1 M KCl, 10 mM MOPS, pH 7.4) under +40 mV bias results in no obvious blockage events (Supplementary Fig. 7). Then, the $Cu^{2+}$ is added as connector because of its capability to form Cys-Cu five-membered ring complex and bind with the Cys126 residue on the C3 channel[34,57]. So, we design to sense L-Cys and L-Hcy with the assistance of $Cu^{2+}$. The mixture solutions of L-Cys (80 μM) and L-Hcy (80 μM) are respectively prepared in the KCl buffer with $Cu^{2+}$ (80 μM). Interestingly, the homogeneous current blockage signals are observed in both L-Cys and L-Hcy mixture solutions at +40 mV (Fig. 2b, c). In detail, the L-Cys leads to smaller current blockage signals ($\Delta I/I_0 = 0.331$) with longer dwell time and the L-Hcy leads to larger current blockage signals ($\Delta I/I_0 = 0.388$) with shorter dwell time (Fig. 2d, e), in which the disparity in $\Delta I/I_0$ for them is attributed to the different sizes of thiol-Cu complexes. The Cys and Hcy events are simultaneously observed in the continuous recording, displaying the distinct discrimination (Supplementary Fig. 8). It is clear that the good resolution between L-Cys and L-Hcy is achieved by our non-engineered ferritin nanopore, although there is only a slight difference by one methylene between them. In addition, multi-level blockage signals are observed both in the detection of L-Cys and L-Hcy, which is considered as the simultaneous sensing for multiple molecules by the multi-channel ferritin sensor (Supplementary Fig. 9a). The multi-level signal of L-Cys, as an example, is analyzed, in which the quantitative relationship between the probability of multi-level signals and the Cys-Cu concentrations is described using binomial distribution model and logistic model (Supplementary Fig. 9b–d). At the same time, compared with ferritin, the apo-ferritin without internal iron core exhibits the identical sensing behavior for Cys-Cu detection, revealing that the conductive and sensing properties of ferritin nanopore are independent of the iron core loaded in ferritin, but attributed to the multi-channel structure and the suitable sensitive sites in the channels (Supplementary Fig. 10).

In addition, to test the qualification capacity of ferritin nanopore, the qualification experiments for L-Cys are carried out. The L-Cys concentration standard curve is fitted according to the acquired event frequency from standard solutions. (Supplementary Fig. 11a) Then, the actual sample (L-cysteine capsule) is dissolved and detected in three independent nanopore experiments. According to the fitted concentration standard curve, we acquired the calculated L-cysteine weight percentage as 16.6 wt%±1.9 wt%, close to the indicated value (18 wt%), revealing the well qualification capacity of the ferritin nanopore (Supplementary Fig. 11b).

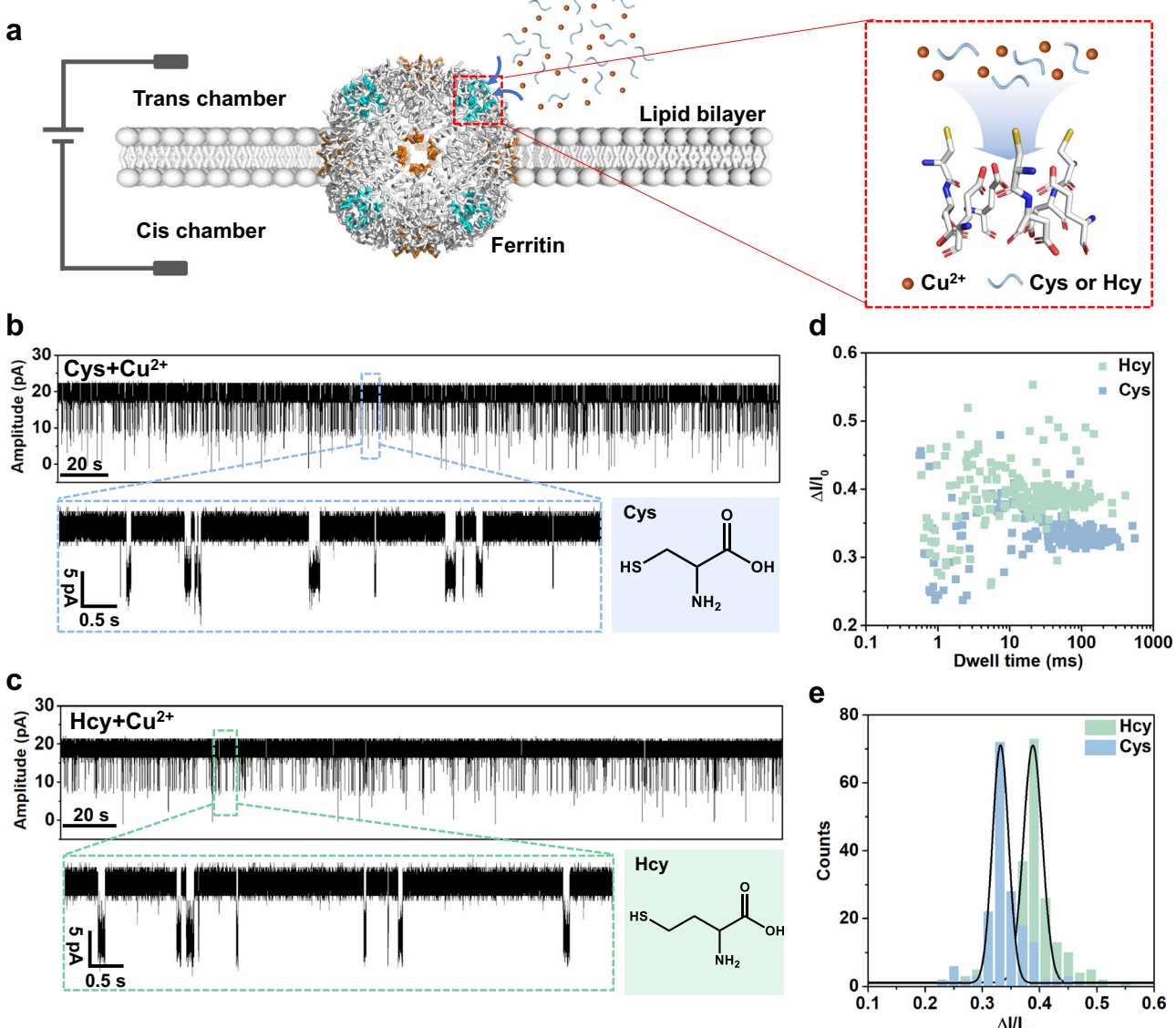

**Fig. 2 | Single-molecule sensing of L-cysteine (Cys) and L-homocysteine (Hcy) by ferritin nanopore with the assistance of Cu²⁺. a** Schematic of ferritin nanopore sensor. The lipid bilayer divides the flow cell into cis (grounded) and trans chambers, and the testing solution is added into the trans chamber for detection. **b** Current trace at +40 mV in the presence of L-Cys (80 µM) and Cu²⁺ (80 µM) in the trans side. **c** Current trace at +40 mV in the presence of L-Hcy (80 µM) and Cu²⁺ (80 µM) in the trans side. **d, e** Scatter plot and histogram showing the dwell time and $\Delta I/I_0$ of blockage signals in Cys ($n = 174$) and Hcy ($n = 201$) sensing.

## Mechanism exploration for cysteine sensing with ferritin nanopore

The key role of Cu²⁺ in the formation of Cys-Cu complex is explored, in which the factors of chelator, buffer, and pH are investigated. The pure Cu²⁺ (80 µM) is firstly tested and there is no obvious signal appearing, indicating the Cu²⁺ itself does not induce blockage signals (Fig. 3a). Then, the key role of Cu²⁺ in the sensing process is further verified by the in-situ nanopore controlled experiments. There are no homogeneous signals appearing with only L-Cys, however, after adding Cu²⁺, the homogeneous blockage events are observed (Fig. 3b). To be noted that, the blockage signals were gradually observed after ~160 s rather than immediate appearance. We guess that in this period, the L-Cys combines with Cu²⁺ forming Cys-Cu complex and causing the characteristic signals. Furthermore, after adding the excess Na₄EDTA (160 µM) into the mixture to capture Cu²⁺, the blockage events disappear, revealing the Cu²⁺ is a key factor in the sensing process (Fig. 3c). The influence of buffer reagents and pH of the testing solution is also evaluated. It is clear that the events frequency decreases

after only replacing the MOPS with Tris-HCl (Fig. 3d, and Supplementary Fig. 12), which is attributed to the strong binding between metal ions and Tris-HCl[58], interfering with the formation of Cys-Cu complex. In addition, the dwell times of blockage signals shorten as pH decreases, which is consistent with the weaker interactions between Cu²⁺ and L-Cys at lower pH (Fig. 3e and Supplementary Fig. 13)[59]. Besides, other metal ions including Fe³⁺, Fe²⁺, Zn²⁺, and Ni²⁺ were also explored as connectors for sensing L-Cys, however, no blockage signals were observed, implying the specificity of Cu²⁺ in this system (Supplementary Fig. 14). All the abovementioned experiments reveal that the coordination between Cu²⁺ and Cys may play the key role in causing blockage events.

To further explore the binding state of Cys-Cu complex, the quantum chemical calculation is used to screen the possible structures, including N-Cu-O, S-Cu-O, and S-Cu-N. The B3LYP-D3/6-31 g(d) method[60,61] and XYGJOS/def2-TZVPP method[62–64] are used to optimize conformations and calculate the relative internal energies of three structures, respectively. As a result, the S-Cu-N five-membered ring

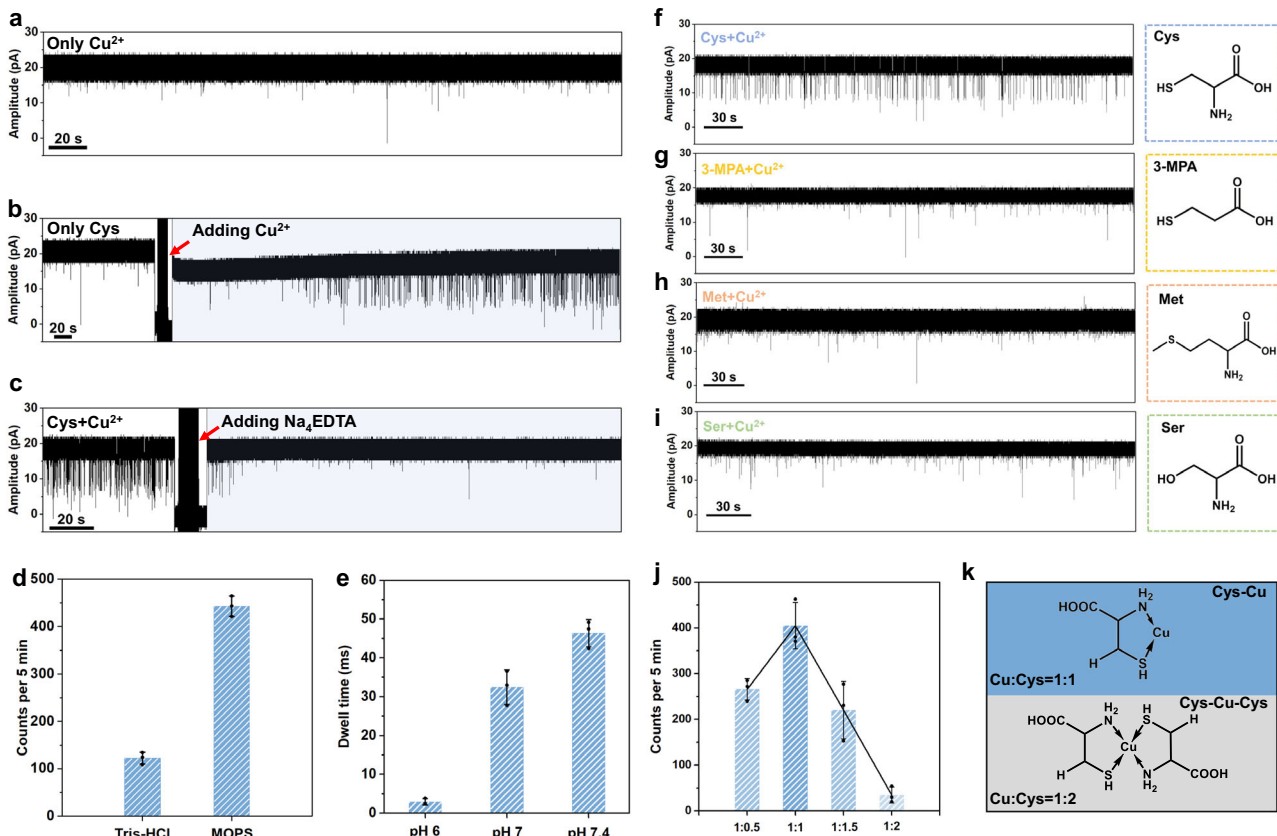

**Fig. 3 | Parameters affecting the events of cysteine analogues during ferritin nanopore sensing. a** Only Cu²⁺ (80 μM) shows no obvious blockage signals.
**b** Current trace of Cys (80 μM) before and after the in-situ addition of Cu²⁺ (80 μM).
**c** Current trace of Cys (80 μM) and Cu²⁺ (80 μM) before and after the in-situ addition of Na₄EDTA (160 μM). **d** Effects of buffers on event frequency of Cys (80 μM) and Cu²⁺ (80 μM). The error bar comes from three different time segments. **e** Influence of pH on the dwell time of Cys (80 μM) and Cu²⁺ (80 μM). The error bar comes from three

independent experiments. **f–i** The controlled experiments for cysteine analogues with different functional groups, showing the essential role of -SH and -NH₂ on cysteine to generate blockage events. **j** The events frequency with the different binding ratios of Cu²⁺ to Cys. The error bar comes from three independent experiments. **k** Proposed structures of Cys-Cu and Cys-Cu-Cys complexes with the different binding ratios of Cu²⁺ to Cys (1:1 and 1:2). For all the bar charts in this figure, the centre and error bar represent mean value and standard deviation (SD), respectively.

binding state is considered as the most stable state for Cys-Cu complex due to its lowest relative internal energy (Supplementary Table 2). In addition, the nanopore controlled experiments are also performed for cysteine analogues, including 3-Mercaptopropionic acid (3-MPA), L-Methionine (L-Met), and L-Serine (L-Ser), which contain only one different functional group comparing to L-Cys or L-Hcy (Fig. 3f-i). As a result, no obvious blockage signals under the same testing conditions were observed for three counterparts, confirming that the synergistic binding of amino (-NH₂) and sulfhydryl (-SH) with Cu²⁺ plays a key role in the formation of S-Cu-N structure to generate characteristic blockage signals. It is worth noting that Cu²⁺ has previously been reported as connectors to help detect amino acids in nanopore methods, where amino acids are thought to bind to Cu²⁺ via their amino and carboxyl groups[9,65]. For example, Zhang et al. utilized MspA nanopore to detect 20 amino acids with the aid of Cu²⁺ [9]. However, the ferritin C3 channel is smaller than the MspA nanopore, leading to a stronger electric field at the entrance. Thus, only the form of most stable S-Cu-N rather than N-Cu-O strongly remains in the channel and generates signals. The sulfhydryl group on analytes is necessary for nanopore sensing with the ferritin nanopore. It is reported that the Cu²⁺ is able to bind one or two cysteine molecules to form different Cys-Cu complexes (Cys-Cu and Cys-Cu-Cys)[57,66]. To explore the influence of the binding ratio of Cu²⁺ to Cys, the solutions with different ratios of Cu²⁺ to Cys (1:0.5, 1:1, 1:1.5, and 1:2) are prepared and tested (Fig. 3j and Supplementary Fig. 15). To be noted that the concentration of Cu²⁺ is fixed at 80 μM, while the concentrations of Cys are set at 40, 80, 120, and 160 μM,

corresponding to the ratios of 1:0.5, 1:1, 1:1.5, and 1:2, respectively. When the ratio of Cu²⁺ to Cys is 1:0.5, the events frequency is calculated to ~265 events per 5 min. After increasing the concentration of L-Cys to the ratio of 1:1, the events frequency improves to ~404 per 5 min. However, with the continued increase of L-Cys, the events frequency starts to decrease and the events are hardly observed with a frequency of ~35 per 5 min in the ratio of 1:2 for Cu²⁺ to L-Cys. According to the experimental results, we speculate that the Cu²⁺ binding with only one cysteine has remaining coordination sites to bind with the residues in the ferritin C3 channel causing the blockage signals, however, the Cu²⁺ chelated by two cysteines has no other coordination sites to interacts with residues in C3 channel resulting in no signals generation (Fig. 3k).

To further explore how the Cys-Cu complex binds with the ferritin C3 channel, a long-time (1 ns) quantum mechanics (QM)/all-atom (AA)/ coarse-grained (CG) MD simulation is carried out. The multiscale model is first constructed. The Cu²⁺, its coordinated residues, and active water molecules are represented by the level of QM with the GFN2-xTB semiempirical method[67,68]. Residues and water molecules near the QM region are defined as the AA region. Other residues and solvents far from the QM region are defined as CG regions. Figure 4a and 4b show the schematic and structural representations of the multiscale model. (Please refer to Methods for more modeling details). To evaluate the reasonability of our multiscale model, the interaction between single Cu²⁺ and residues in the C3 channel is simulated and compared to that in the crystal structure 3RE7[34,69]. The observed coordination pattern of Cu²⁺ with His114, Cys126, and Glu130 in the

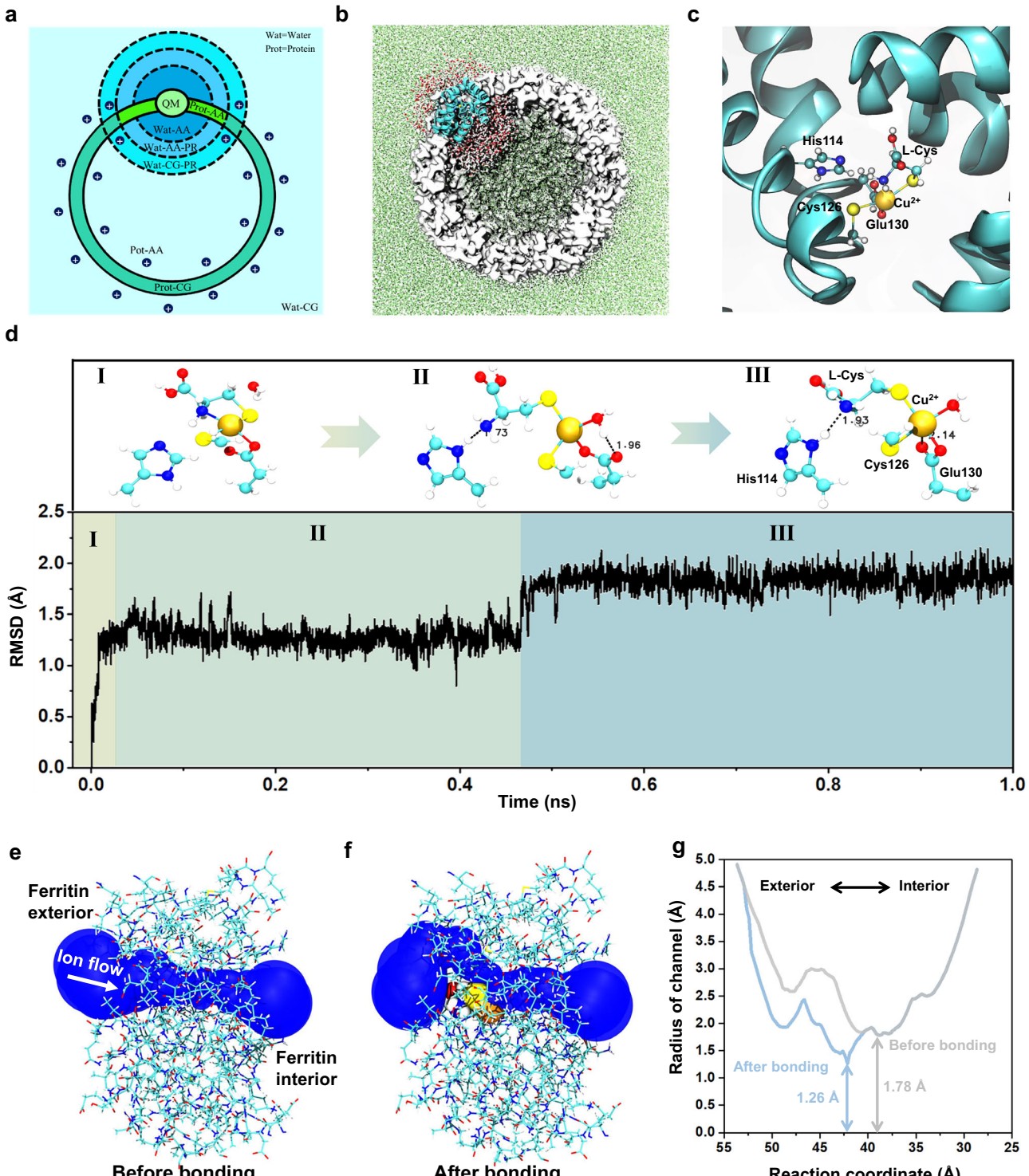

**Fig. 4 | Simulation of binding conformations of Cys-Cu-ferritin based on the multiscale (QM/AA/CG) model. a** A schematic diagram of the CG division map for different regions. Cys-Cu-Ferritin (Prot) is represented by three scales, QM, AA, and CG. Water solvents (Wat) are represented by two scales, AA and CG. Position restrictions (PR) are adopted for molecules near interfaces to prevent diffusion. **b** The initial conformation of the active and non-active region in the Cys-Cu-Ferritin. The active region corresponds to the AA & QM regions in **a** represented by the cyan cartoon (protein) and AA water molecules. Non-active CG regions are depicted using white surfaces (protein) and green dots (solvents), respectively. **c** The initial conformation of the QM region represented by Cu²⁺, L-Cys, His114,

Cys126, and Glu130 and two nearby water molecules. **d** The root mean square deviation (RMSD) of Cu²⁺ and five coordinated atoms from residues with respect to the simulation time. Primary coordination patterns of corresponding RMSD are given above, with hydrogen bonds represented by doted lines. **e, f** The side view showing the shape of the ion transferring pore before and after Cys-Cu attached to the C3 channel. **g** The estimated radius of the channel before and after bonding Cys-Cu complex with respect to the reaction coordinate of ion flow. The reaction coordinate indicates the position of the ion flow within the C3 channel, with larger values corresponding to locations nearer to the exterior of the ferritin.

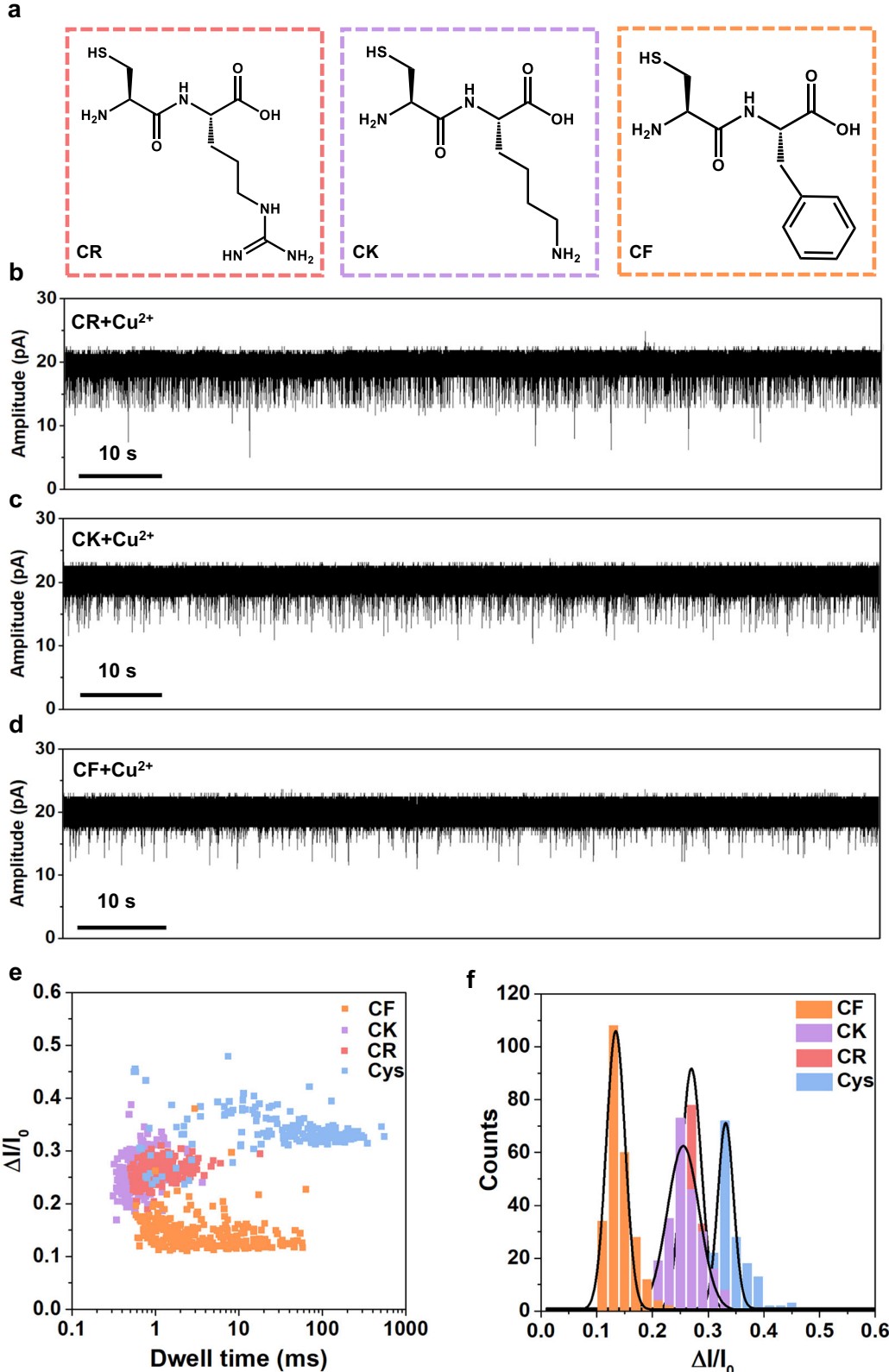

**Fig. 5 | Single-molecule sensing of cysteine-containing dipeptides by ferritin nanopore with the assistance of Cu²⁺. a** Structures of CR, CK, and CF dipeptides. **b–d** Current traces at +40 mV of CR (80 µM), CK (80 µM), and CF (80 µM) in the presence of Cu²⁺ (80 µM), respectively. **e, f** Scatter plot and histogram showing the dwell time and $\Delta I/I_0$ of blockage signals in dipeptides detection. The signals of cysteine are also displayed for comparison. $n = 174$, 214, 238, and 251 for Cys, CR, CK, and CF, respectively.

simulation is similar to that in the crystal structure, confirming the reasonability of our multiscale model (Supplementary Fig. 17).

Then, the binding state of Cys-Cu complex in C3 channel is explored with QM/AA/CG multiscale MD simulation. First, the Cys-Cu complex is docked into the protein channel in a manner similar to the Cu(II) coordination in the crystal structure 3RE7. Cu(II) adopts a square pyramidal coordination mode, with two sulfur atoms in a trans position to reduce steric hindrance, the coordination pattern is shown in Fig. 4c. Afterwards, the Cys-Cu-Ferritin complex is solvated and coarse-grained to the QM/AA/CG multiscale system. The MD simulation is carried out for 1 ns to evaluate the stability of the coordination. The coordination pattern of the Cys-Cu-ferritin complex undergoes two major transitions (step I to II and step II to III) during the simulation, as shown in Fig. 4d and Supplementary Fig. 18. The coordination between -SH on Cys126 and Cu(II) always remains stable and the two oxygen atoms from -COOH on Glu130 alternately bind to the Cu(II), indicating both the Cys126 and Glu130 on the channel have stable interaction with Cu(II). As for the L-Cys, its -SH keeps stable interaction with Cu(II) but the -NH$_2$ gradually moves away from Cu(II) and forms a stable hydrogen bond with the imidazole group on His114. In this way, the Cys-Cu complex is captured and stabilized in the C3 channel with multiple interactions from residues of His114, Cys126, and Glu130. To further verify the key roles of these residues, we introduce excess Zn$^{2+}$, that also has stronger interaction with these residues[34], in the detection for Cys-Cu complex to occupy these binding sites and analyze how the Cys-Cu events frequency becomes before and after introducing Zn$^{2+}$. The MD simulation was first conducted to further confirm the coordination style of Zn$^{2+}$ in C3 channel, in which, the Zn$^{2+}$ was observed to indeed bind with these three residues and keep stable (Supplementary Fig. 19a, b). Furthermore, the concentration control experiments for Zn$^{2+}$ were conducted, and it was found that the frequency of Cys-Cu events gradually decreased as the Zn$^{2+}$ concentration increased, accordingly revealing the key roles of these residues (His114, Cys126, and Glu130) in binding with Cys-Cu complex (Supplementary Fig. 19c). In addition, the C3 channel radiuses before and after bonding Cys-Cu complex are calculated by MolAICal software[70] (Fig. 4e–g). After bonding Cys-Cu complex, the channel radius at the narrowest point shows an obvious reduction, which causes the smaller ionic current, indicating the Cys-Cu complex is the key analyte in causing current blockage signals. Combining with the above controlled nanopore experiments and MD simulations, we result that the -NH$_2$ and -SH of L-Cys play a key synergistic effect in chelating Cu$^{2+}$ to form Cys-Cu complex. The Cys-Cu complex will enter into the ferritin C3 channel and then bind with His114, Cys126, and Glu130, causing the blockage events.

## Dipeptides sensing with ferritin nanopore

To further explore the sensing applications, we attempt to detect cysteine-containing dipeptides with the ferritin nanopore. Five dipeptides including Cys-Phe (CF), Cys-Arg (CR), Cys-Lys (CK), Cys-Asp (CD), Cys-Glu (CE) are detected and we find that CF, CR and CK display homogeneous blockage signals while CD and CE cause no signals (Fig. 5a–d and Supplementary Fig. 20). Negatively charged molecules, as reported by Yang et al., are hard to enter the cavity of ferritin via negatively charged C3 channel due to electrostatic repulsion[71], which can explain why the negatively charged CD and CE cause no signals (Supplementary Fig. 21). The signal statistics reveal that all CF, CR, and CK have larger volumes but smaller blockage fractions than L-Cys (Fig. 5e, f). As reported, all the charge, polarity and size of analytes can influence the current blockage fraction in nanopore analysis[6,9,10]. In our experiments, the blockage fractions of CK, CR, and CF are all smaller than that of L-Cys, with CF having the smallest $\Delta I/I_0$, and the difference between CK and CR is minimal (Fig. 5e, f). It is understood that the smaller L-Cys is able to enter into the deeper site of the channel causing larger blockage fraction, while the dipeptides have too large

volumes to enter deeper, causing smaller blockage fractions. The CF has the strongest hydrophobicity and largest volume in all of them displaying the smallest blockage fraction. Given that lysine (K) and arginine (R) are hydrophilic polar amino acids, CK and CR are more likely to penetrate deeper into the C3 channel compared to CF, resulting in larger blockage fractions than CF. The charge interaction is believed to dominate in such a strongly negatively charged C3 channel, making it difficult to distinguish between the similarly positively charged CK and CR despite their difference in volume. To decrease the charge interaction between CR/CK and C3 channel, we improve ionic strength by using higher concentration KCl buffer (2 M KCl) in nanopore experiments (Supplementary Fig. 22). In this condition, the blockage fraction is dominated by the volume exclusion effect, which leads to a larger $\Delta I/I_0$ for larger CR.

In summary, we have constructed a new type of nanopore with ferritin by inserting it into the lipid bilayer membrane. The ionic current recordings and multiscale MD simulations reveal that the ferritin can be embedded into the lipid bilayer and stably maintained, in which the presence of multiple C4 hydrophobic channels plays a crucial role in stabilizing the ferritin. The asymmetric *I-V* curves and channel opening/closing behavior under high voltage were discovered in single-ferritin electrophysiological recordings, revealing the ionic current rectification and voltage-gating characteristics of the ferritin nanopore. We then use the ferritin as a novel nanopore sensor to achieve the detection and differentiation of L-Cys, L-Hcy, and cysteine-containing dipeptides with the assistance of Cu$^{2+}$, which demonstrates the high resolution of ferritin nanopore. The detection mechanism was also explored by combining the nanopore experiments and MD simulation using cysteine as an example. The cysteine combines with Cu$^{2+}$ to form Cys-Cu complex and then the complex is captured by the ferritin C3 channel with multiple interactions from His114, Cys126, and Glu130, causing the current blockage signals. The ferritin nanopore provides a high resolution sensing platform for single-molecule detection. Moreover, the successful construction of the non-barrel-shaped ferritin nanopore provides a new perspective on screening and designing advanced protein nanopores.

## Methods
### Materials
1,2-diphytanoyl-sn-glycero-3-phosphocholine (DPhPC) was from Avanti Polar Lipids and *n*-decane was from Aladdin (China). Tris (hydroxymethyl)aminomethane hydrochloride (Tris-HCl), potassium chloride (KCl), and horse spleen ferritin (F4503, SLCJ8967) were obtained from Sigma-Aldrich (China). 2-Morpholinoethanesulphonic acid (MES), 3-Morpholinopropanesulfonic acid (MOPS), potassium hydroxide (KOH), copper chloride (CuCl$_2$), L-cysteine (L-Cys), L-homocysteine (L-Hcy), 3-Mercaptopropionic acid (3-MPA), and L-Methionine (L-Met) were from Aladdin (China). L-serine (L-Ser) was from Bidepharm (China). Hydrochloric acid (HCl) and ethanol (C$_2$H$_5$OH) were from Sinopharm (China). All the dipeptides were obtained from Sangon (Shanghai). All solutions in nanopore experiments were prepared with Wahaha pure water. The CuCl$_2$, L-Cys, L-Hcy, L-Ser, L-Met, 3-MPA, and all the dipeptides were respectively dispersed into Wahaha pure water to prepare the stock solutions in 8 mM. Those stock solutions would mix with KCl buffer (pH 7.4, 1 M KCl, 10 mM MOPS) and be diluted into proper concentration for nanopore experiments. The ferritin solution was prepared by pipetting 5 μL ferritin sample (65 mg/mL) into 1 mL KCl buffer.

### Preparation of ferritin nanopore sensing apparatus
1 mg DPhPC is dispersed into 100 μL n-decane to make the membrane-forming solution (10 mg/mL). 1 μL lipid solution is pipetted onto the hole of the Delrin perfusion cup (Warner Instruments, cat. no. 64-0423) for 15 min to naturally vaporize the *n*-decane. After that, the cup is assembled as a flow cell and 1 mL electrolyte solution is pipetted into

each chamber of the flow cell. 1 μL lipid solution is pipetted into the cis chamber and the solution interface is constantly lowered and raised by taking and releasing the solution with a pipette to form the lipid bilayer on the hole. A reliable lipid bilayer membrane can keep stable under +180 mV and come to a rupture over +300 mV. The ferritin solution is pipetted into the cis chamber near the lipid bilayer membrane and +180 mV bias is applied to promote the insertion of ferritin molecule. A current change from 0 to about 150 pA implies the successful insertion of a single ferritin and then the bias is reset to +40 mV for nanopore experiments.

### Data acquiring and analysis

All the single-molecule signals are acquired with an Axopatch 200B patch clamp amplifier and an Axon Digidata 1550B digitizer (Molecule Device, USA). All the signals are recorded with 100 kHz sampling rate and filtered through a 1 kHz lowpass Bessel filter. Unless otherwise noted, all electric recordings are carried out with +40 mV at room temperature (25 °C) and 80 μM $Cu^{2+}$ is present in trans chamber for all the nanopore experiments to assist the detection of thiol-based analytes. All events are searched with the "single channel search" method built in the Clampfit 11.1. All the scatter plots, histograms and curve fitting are generated with Origin 2016.

### Coarse-grained and multiscale MD simulation

(1).  AA and CG models for residues, lipid bilayer, ligands and solvents. The all-atom (AA) and coarse-grained (CG) models of small molecules and residues are modeled in a general way and subsequently used to construct complex systems. The AA residues, ligands (including $Cu^{2+}$ and L-Cys), and solvents are represented by AMBER-ff19ipq, GAFF (with RESP charges), and TIP3P models respectively[72–75]. The CG residues and lipids are constructed systematically from bottom-up coarse-graining approaches. The CG division (not forcefields) of residues and lipids are defined the same as the MARTINI model and the united-atom model[76]. Their non-bonded interactions, which include Lennard-Jones (LJ) interactions and Coulomb interactions, are parameterized consistently based on the Lennard-Jones Static Potential Matching (LJSPM) and the AA member charge summation[46,77]. The bonded interactions are written in an AMBER-liked potential energy function[74,75], but parameterized in a hybrid approach, where the inter-residue and inter-lipid parameters are deduced based on the direct Boltzmann inversion[78] and the peptide bonds of AA-CG or CG-CG residue pairs are parameterized by the structure-based approach. The CG water molecules are described by two kinds of CG particles with equal and opposite charges ($\pm 0.40$), and identical LJ parameters as the OW atom in TIP3P. This ionic-liquid-liked CG water can well match critical physical properties like density, diffusions.

(2).  Construction and MD simulation of ferritin-lipid bilayer system. The ferritin-lipid bilayer CG model is used to evaluate whether ferritin can stably exist within the lipid bilayer. The ferritin model is constructed based on the reported PDB structure (PDB: 2W0O)[79] and all the residues of ferritin in this simulation system are described by the CG resolution. The system is neutralized by evenly distributing $K^+$ ions inside and outside the ferritin cavity. The elastic network model (ENM) with a force constant of 10.0 kcal/Å² is used to maintain the secondary conformation of backbones[80]. The lipid and water models are also described by the CG resolution. The lipid bilayer is composed by total 800 phospholipid molecules, 400 for the upper and 400 for the lower layer. The CG water molecules are equally distributed to the whole system (including the cavity) by density.

The MD simulation for ferritin-lipid bilayer system is carried out with Gromacs-2023.1 software[81] and the simulation duration is set to 6 μs with a step of 6 fs. The charge interactions are calculated with Particle Mesh Ewald (PME) method. The temperature and pressure are respectively controlled at 373 K and 1 bar with v-rescale and c-rescale methods and the cutoff is set to 1.2 nm.

(3).  MD simulation of the interaction between $Cu^{2+}$ and residues in ferritin C3 channel. The QM/AA/CG multiscale ferritin model is constructed to verify the combination mode of $Cu^{2+}$ in the ferritin C3 channel. To correctly describe the combination mode of $Cu^{2+}$, molecules (including residues and solvents) nearby $Cu^{2+}$ are described by the quantum mechanics (QM) model, through which the electronic effect can be correctly taken into account. The QM region is defined as $Cu^{2+}$, three possible coordinate residues (His114, Cys126, Glu130) and two active water molecules. The QM Hamiltonian is computed by the GFN2-xTB semiempirical method, which has been previously utilized to describe systems containing Cu(II)[67,68].

Molecules within a cutoff range (20.0 Å for residues and 30.0 Å for water) from the QM region are all described by the AA resolution to accurately simulate short and middle range environment effects for the QM region. These are denoted as the Port-AA, Wat-AA regions. The position restraint (PR) is applied to the outer shell of the AA solvent sphere (solvents outside 25.0 Å) to ensure that they do not escape to the CG region, denoted as the region Wat-AA-PR.

Other molecules are described by the CG resolution to efficiently reproduce the long-range environment effects, including CG residues and solvents (Prot-CG, Wat-CG). PR is also applied to Wat-CG within 70.0 Å from the QM region to ensure that they do not enter into the AA or QM region. Meanwhile, ENM is used to maintain the secondary conformation of backbones. The force constant for both PR and ENM is defined as 10.0 kcal/Å².

The MD simulation is carried by the AMBER 23, ORCA 5.0.4, and xTB software for 1 ns with the timestep of 1 fs[68,82–84]. The Lennard-Jones cutoff distance is set to 12.0 Å. The QM/MM interactions are described by the electronic embedding method with the qm_cut set to 20.0 Å. The temperature and pressure are respectively controlled at 300 K and 1 bar with the v-rescale and Berendsen method.

(4).  MD simulation of the interaction between Cys-Cu complex and residues in ferritin C3 channel. The modeling and the simulation details are almost identical to that of the QM/AA/CG simulation mentioned above. The only difference is that the QM region additionally includes the Cys molecule coordinated with $Cu^{2+}$.

### Reporting summary

Further information on research design is available in the Nature Portfolio Reporting Summary linked to this article.

## Data availability

The data that support the conclusions of this study are either presented in the paper or its Supplementary Information. Source data are provided with this paper. Source data is available for Fig. 1e–h, Fig. 2, Fig. 3d–j, Fig. 4d, g, Fig. 5 and Supplementary Fig. 1b. Source data are provided with this paper.

## Code availability

The python code for binomial distribution model is available. Code files have been deposited in Figshare. Please follow the link https://doi.org/10.6084/m9.figshare.28953416.v1 for download.

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

## Acknowledgements

This work is supported by the Jiangsu Basic Research Center for Synthetic Biology (Grant No. BK20233003, Z.-Y.G.), the National Natural Science Foundation of China (22174067 and 22374077, Z.-Y.G.;

22403051, Y.-W.Z.), and the Priority Academic Program Development of Jiangsu Higher Education Institutions (Z.-Y.G.). We thank Dr. Ling-Yu Chu and Prof. Guan-Zhu Han for their kind assistance with the SDS- and Native-PAGE experiments. We thank Prof. Ke-Long Fan for his friendly providing recombinantly expressed and purified human L-ferritin (LFn) and H-ferritin (HFn).

## Author contributions

Z.-Y.G. conceived the idea and supervised the research. Y.-D.Y. and X.-T.S. performed the nanopore experiments and discussed the results. Y.-W.Z. and Y.-F.L. performed the MD simulation. J.H. assisted in the controlled experiments. Y.-H.C. assisted in the MD simulation. Y.-D.Y. wrote the paper. Y.-D.Y., Y.-W.Z., W.-C.L., and Z.-Y.G. discussed the experimental data and revised the paper. All authors have approved the final version of the manuscript.

## Competing interests

The authors declare no competing interests.
