## [Transparent Peer Review file · Nature Communications]

Native Globular Ferritin Nanopore Sensor

Corresponding Author: Professor Zhi-Yuan Gu

Version 0:

Reviewer comments:

Reviewer #1

(Remarks to the Author)

This manuscript by Gu et al has reported the use of Ferritin as a new type of nanopore sensor, with which two biothiols including cysteine and homocysteine has been detected. They further demonstrated ferritin sensing of a few dipeptides including cysteine residues. Some MD simulation and quantum chemistry calculations were also carried out to support their conclusions. Generally, I am a bit surprised that a natural ferritin protein may be directly used as a nanopore sensor. Besides, the channel stability and its small molecule sensing capacity are both quite good. The manuscript does contain some interesting new phenomenon and novelty. However, the manuscript in its current form is not yet ready for this journal. Actually, I found this newly discovered nanopore not quite advantageous and plenty of issues exist or unclarified at this stage as well. I am listing them below for the ease of your reference.

1. A clear description of this newly discovered nanopore. As stated by the authors, Ferritin is composed of 24 subunits. The authors then mentioned that the same ferritin contains 6 C4 channels and 8 C3 channels. Each C3 channel contains 3 natural cysteine residues. Does that mean a single ferritin contains a total of 24 natural cysteine? Since this is the first report of ferritin as a nanopore sensor, it is important to clearly describe its structural properties as a pore. According to the current description, I am quite confused what kind of nanopore it is. It contains multiple channel? Are they simultaneously activated when the protein is inserted in the membrane? How many metal binding sites do a single ferritin have? It is also not very clear according to the schematics in Fig 1. Is the ion transporting through the C3 or the C4 channel? Or their combination? The authors should clarify these facts before acclaiming it as a nanopore sensor. I guess a cross section demonstration of the pore inserted in the membrane, just like most other nanopores, would help.
2. Clearly describe the origin of the ferritin sample at its first appearance. Was it ordered from a company or the team prepared it all by themselves? These information are important for others to replicate their results. Briefly introduce them in the maintext.
3. MD simulation. It is appreciated to have MD simulation results. However, I don't quite get what kind of conclusion did you draw from the simulation? It seems that no valuable information was obtained from the simulation? Then why do we need the simulation at all?
4. It is called a ferritin, does it naturally contain any bound Fe²⁺ or Fe³⁺ ion in the protein at all? The authors then used Cu²⁺ to realize biothiol sensing. Why use Cu²⁺ if this protein seems to be a Fe²⁺ or Fe³⁺ binding protein?
5. Though I am not 100% sure, it seems that there are multiple equivalent Cu²⁺ binding sites on the same ferritin nanopore. If so, why don't we observe simultaneous binding of multiple Cu²⁺ and Cys/Hcy binding events.
6. Simultaneous discrimination of Hcy and Cys using the same pore. To acclaim nanopore discrimination of Hcy and Cys, the authors should provide simultaneous discrimination of Hcy and Cys using the same nanopore sensor and show a continuous trace lasting for a few mins. If this nanopore sensor does have the resolution to discriminate between Cys and Hcy, then two types of events should be alternatively observed in the same continuous trace. The corresponding event scatter plot results should also show two clear populations. Please note that this demonstration should not be skipped.
7. Previous work demonstrating nanopore discrimination of Cys and Hcy. It seems that the authors fail to cite any previous reference of nanopore discrimination of Cys and Hcy (Chem. Commun., 2019,55, 9311-9314; Nat Commun 10, 5668 (2019).) The authors also acclaimed that "the discrimination between L-Cys and L-Hcy has not been previously reported with non-engineered nanopore tools". I guess no one cares whether a nanopore is engineered or not. People care whether this sensing task has been previously done, how it was done and how well it has been done. So it is more appropriate to cite relevant works, describe the concept difference between your idea and theirs. Compare their pros and cons.
8. As recently reported, a nanopore bound with Cu²⁺ ion is generally sensitive to all amino acids (Nat Methods 21, 609–618 (2024).). Another earlier study of Cu²⁺ nanopore for amino acid sensing was found as well (Angew. Chem. Int. Ed., 51: 9606-9609). If that is the case, why do you observe binding of thiol containing amino acids? I am quite shocked that the

Angew Chemie paper has not been cited at all. The Nat Methods paper was cited but not discussed and compared with this technique at all?

9. Quantification issue. Being a Cys/Hcy sensor, it seems that at a higher concentration of Cys/Hcy, the rate of event appearance is actually dropping. If that is the case, it means that this nanopore sensor has a big problem to deal with quantification of Cys/Hcy, which is quite disadvantageous.

10. Fig 5, it seems that the discrimination between CK and CR seems to be not achieved? Does that mean the nanopore can only discriminate between CF and CK or CR? If that is true, it means that the nanopore resolution seems to be disadvantageous. I suggest removing the event populations of CK to SI. Leave only events that are clearly discriminated in the main text figure.

11. Explain why CD and CK cause no signal at all?

12. The pore is noisy when a high voltage is applied, meaning that it can only be used at a low applied bias, such as 40 mV. While most biological nanopores can withstand 200 mV bias without a problem. This means that this ferritin pore is disadvantageous in the resolution.

Eventually, based on all above discussions, though I found this newly discovered ferritin nanopore quite interesting and novel, it is generally disadvantageous as a nanopore sensor by having a low resolution, an unclear structure and binding site for future engineering. Plenty of grammar and typo issues exist as well. The authors also failed to cite strongly relevant previous works with similar concepts. It is always good to new types of nanopores, good for the field. However, to qualify publication at a high impact journal like Nature Communications, this new nanopore should show its unique sensing properties and advantages. I hope my above suggestions may help the authors to further improve this manuscript.

Reviewer #2

(Remarks to the Author)

Nanopore sensing, renowned for its label-free, highly sensitive, accurate, cost-effective, high-throughput, and portable characteristics, has garnered significant attention in recent years. A key area of focus is the development of new nanopores with enhanced stability and spatial resolution. Through a combination of experimental results and molecular dynamics (MD) simulations, the author provides compelling evidence that a single ferritin molecule can be inserted into a lipid bilayer, forming a stable ionic channel. Further exploration demonstrated the sensing capabilities for cysteine, homocysteine, and cysteine-containing dipeptides. High-quality experimental data and computational simulations elucidated the mechanism by which the ferritin nanopore detects these molecules, facilitated by Cu^{2+} . This study underscores the potential of ferritin for ultrasensitive single-molecule sensing, given its pore size of 3-5 Å. However, the ability to identify dipeptides has not been fully explored, and the validation of some hypotheses remains insufficient. Further revisions are needed to clarify the methodology and its robustness.

Here are some specific points:

1. It is somewhat counterintuitive that a spherical protein could be stabilized within a lipid bilayer. Is this an isolated example, or is it possible to find more spherical proteins that can be inserted into lipid bilayers?

2. The scatter plot of CK and CR nearly overlaps. Can these two dipeptides be differentiated under altered experimental conditions?

3. The underlying mechanism behind the absence of signals from CD and CE should be further investigated. There is insufficient evidence to support the conjecture that the presence of COOH groups may prevent peptides from binding to nanopores. The negative charge may affect the translocation of the dipeptide. Additional supporting experiments or simulations should be conducted.

4. The author claimed that "the smaller L-Cys is able to enter into the deeper site of the channel causing a larger blockage fraction." However, this claim is contradicted by the observation that R, which has a larger volume than K, shows a deeper blockage in the CR dipeptide.

Reviewer #3

(Remarks to the Author)

In their manuscript, Yun-Dong et al. describe the application of horse spleen ferritin (a mostly L-ferritin oligomer) as a nanopore bio-sensing analytical device.

The authors use single-channel voltage clamp recordings to demonstrate that ferritin can function as a stable trans-membrane ion channel. The authors combine their experimental work with computational simulations to provide insights into the molecular mechanism of transport through the ferritin channels. The authors test the ferritin nano-pore for the detection of amino acids, using the binding of copper ions to enhance the sensing capabilities. They then use an elaborate analysis of the system by adding/changing different parts of the system (e.g., Cu ions, buffers) to elucidate the molecular basis of the sensing.

Overall, this research is innovative, with compelling results and discussion; thus, I strongly recommend its publication.

However, there are several weaknesses/shortcomings that I recommend addressing before publication:

The treatment of the protein sample is too superficial. Horse spleen ferritin is mostly L-ferritin but not only L-ferritin. Thus, the authors should at least characterize the sample used (e.g., SDS and Native PAGE, size-exclusion chromatography, etc.) to determine the actual composition. Preferably, they should repeat at least one experiment with a recombinantly expressed

and purified protein (L-ferritin is easily purified).

If the authors' claims are correct, the system has four pores above and below the membrane. In such a case, one can calculate the number of events where more than one pore is blocked simultaneously and how the propensity of these events changes with the concentration of analytes. Can you find that in your data?

The role of three amino acids in the pore is discussed in detail. I recommend testing these claims by repeating at least one experiment with a mutant of L-ferritin replacing one (or more) of these amino acids.

While the paper is easily read and well organized, some annoying grammatical errors (e.g., problems with plural/singular messing "the" etc.) should be addressed.

The term "proven" is used too lightly and should be avoided.

Version 1:

Reviewer comments:

Reviewer #1

(Remarks to the Author)

The authors have successfully addressed all my review comments. The manuscript is now ready to be published.

Reviewer #2

(Remarks to the Author)

The authors have addressed all the concerns raised during the first-round reviewing. In general, the work is carefully executed and the results are quite interesting. It may prompt researchers in the field to test many other proteins like ferritin as nanopores.

Reviewer #4

(Remarks to the Author)

I have significant concerns regarding the use of horse spleen ferritin (purchased from Sigma-Aldrich) in this study, as it introduces multiple uncertainties that could compromise the validity and reproducibility of the results. First, the ferritin used appears to be in its holo-form (iron-loaded), given the information provided in the authors' response. Horse spleen ferritin can contain thousands of iron atoms, which could significantly influence the nanopore behavior and signal responses, particularly in experiments intended to study specific interactions with other metals like copper. The observed lack of blockage signals could be attributable to the overwhelming presence of iron rather than the intrinsic properties of the ferritin nanopore system. Furthermore, the choice of horse spleen ferritin without further purification raises additional issues of purity and consistency. Based on prior experience, commercially available horse spleen ferritin from Sigma-Aldrich often lacks sufficient purity for rigorous scientific studies and can yield inconsistent data. This raises questions about why the authors did not opt for higher-quality alternatives, such as human recombinant ferritin, which would offer better control over subunit composition (H or L), iron loading, and overall reproducibility. Thus, I can fully understand the concern raised by Reviewer 3. My observations and feedback on the authors' responses to Reviewer 3 are outlined below:

Question 1

The authors have not adequately addressed the first question raised by Reviewer 3. Instead, they appear to have taken a simplistic approach by presenting gel data (SDS and PAGE) without providing comprehensive characterization of the horse spleen ferritin composition, as requested. Additionally, the authors did not repeat the experiments using recombinantly expressed and purified ferritin proteins, such as homopolymer L-ferritin or H-ferritin, which are widely utilized in the field. It remains unclear whether the reported nanopore sensing capability attributed to ferritin is also observed with homopolymer H-ferritin or heteropolymer ferritins composed of varying combinations of H and L subunits. Additionally, the effect of a large iron core present in horse spleen ferritin on the sensing properties of ferritin has not been explored and been largely ignored in these experiments.

Question 2

The authors have partially addressed the concern regarding the multi-level signals and the relationship between analyte concentration and simultaneous pore blockages (Question #2). They provided data showing an increase in the proportion of multi-level blockage signals with higher concentrations of L-Cys (Supplementary Fig. 9). However, the original concern is about the calculation and analysis of multi-level events (i.e., simultaneous pore blockages) and how these events vary with analyte concentration. The Reviewer was interested in a quantitative approach to understand and predict the likelihood of multi-level signals. This addition does not address the statistical analysis or propensity of simultaneous blockages. The added experiment (Supplementary Fig. 10) does not address the frequency or behavior of multi-level signals explicitly, which was the core of the question. Additionally, the new data does not provide insights into the mechanistic basis of multi-level blockages or explain the multi-channel behavior of the ferritin nanopore.

Question 3

Once again, the authors have sidestepped the original question and opted for an alternative approach that raises additional concerns. First, the authors cite a lack of time and expertise in bioengineering as reasons for not preparing an L-mutant ferritin. However, there are numerous bioengineering companies and suppliers capable of providing mutant ferritin within a relatively short timeframe. Second, the alternative approach that the authors introduced employed a large excess of zinc

(0.24 mM), but it remains unclear where the added zinc binds—whether in the C3 channels or on the ferritin surface. The authors have assumed, without supporting evidence, that the zinc ions would replace the copper ions in the Cys-Cu complexes within the C3 channels. Thus, this experiment failed to address the reviewer's concern and to conclusively demonstrate the critical role of the three amino acids (His114, Cys126, and Glu130) in stabilizing the Cys-Cu complexes.

Version 2:

Reviewer comments:

Reviewer #4

(Remarks to the Author)

The authors have thoroughly and satisfactorily addressed my questions and concerns, providing additional data and detailed explanations that have strengthened the study and made the manuscript's conclusions more robust. Although both HF_n and LF_n share the key coordinating residues in the C3 channel (His114, Cys126, and Glu130), differences in their overall structure and electrostatic or surface properties appear to have prevented HF_n from embedding in the membrane and functioning as a nanopore under the tested conditions. This is likely due to variations in surface charge and hydrophobicity, which are critical for membrane insertion and nanopore activity. I recommend that the authors include this information in the main manuscript for clarity. This result represents a novel and important finding that emerged only after the second round of reviews.

I would also encourage the authors to consider investigating a range of H/L subunit compositions (in a future study) to further optimize sensor performance. A blend of H and L subunits could potentially provide improved properties, such as enhanced membrane embedding, stability, or selectivity, compared to L-rich ferritin alone, offering valuable insights into how subunit composition affects nanopore behavior and analyte recognition. This is particularly interesting since physiological ferritin is typically a heteropolymer with subunit composition relevant to numerous disease states (e.g. iron metabolism disorders, cancer, and neurodegeneration). Such research is feasible given the recent advances in synthesizing and purifying a range of H/L subunit compositions and could help bridge fundamental nanopore science with clinical applications, making these findings more broadly impactful and translational.

Response to Reviewers' Comments

Reviewer: 1

Comments: This manuscript by Gu et al has reported the use of Ferritin as a new type of nanopore sensor; with which two biothiols including cysteine and homocysteine has been detected. They further demonstrated ferritin sensing of a few dipeptides including cysteine residues. Some MD simulation and quantum chemistry calculations were also carried out to support their conclusions. Generally, I am a bit surprised that a natural ferritin protein may be directly used as a nanopore sensor. Besides, the channel stability and its small molecule sensing capacity are both quite good. The manuscript does contain some interesting new phenomenon and novelty. However, the manuscript in its current form is not yet ready for this journal. Actually, I found this newly discovered nanopore not quite advantageous and plenty of issues exist or unclarified at this stage as well. I am listing them below for the ease of your reference.

Response: Thank you so much for your time, effort, and expertise in reviewing our work. Thanks for your recognition of our novelty. Your mentioned questions have been responded as follows.

1. A clear description of this newly discovered nanopore. As stated by the authors, Ferritin is composed of 24 subunits. The authors then mentioned that the same ferritin contains 6 C4 channels and 8 C3 channels. Each C3 channel contains 3 natural cysteine residues. Does that mean a single ferritin contains a total of 24 natural cysteine? Since this is the first report of ferritin as a nanopore sensor, it is important to clearly describe its structural properties as a pore. According to the current description, I am quite confused what kind of nanopore it is. It contains multiple channel? Are they simultaneously activated when the protein is inserted in the membrane? How many metal binding sites do a single ferritin have? It is also not very clear according to the schematics in Fig 1. Is the ion transporting through the C3 or the C4 channel? Or their combination? The authors should clarify these facts before acclaiming it as a nanopore sensor. I guess a cross section demonstration of the pore inserted in the membrane, just like most other nanopores, would help.

Response 1: Thank you so much for your constructive suggestions. There are eight C3 channels, composed of three-folds of Cys126, Asp 127 and Glu 130 residues, on a single ferritin. As you understand, a single ferritin contains a total of 24 natural cysteine in eight C3 channels. As shown in **Supplementary Fig. 2 and Fig. 2a**, the ferritin embeds in the lipid bilayer with four C3 channels on

each side, constructing a multiple channels nanopore. All the C3 channels are considered to be equivalent and simultaneously activate when the protein is inserted in the membrane. The observed multiple-level events (please see question 5 for details) can support that all the C3 channels are simultaneously active. Every C3 channel is composed of three-folds of Cys126, Asp 127 and Glu 130 residues. According to the MD simulation, the Cys 126 and Glu 130 play the key role in synergistically binding one Cu (II). (Fig. 4c) Thus, there are 3 Cu (II) binding sites on one C3 channel and totally 24 sites on a ferritin. However, we consider only one Cu-Cys complex can be simultaneously captured by the C3 channel due to the large steric effect in such a narrow channel. As previously reported (*PNAS*, 2014, **111**, 7925-7930), the C3 channel as the iron transport path is hydrophilic and the C4 channel is hydrophobic. Thus, the ions can only transport through the C3 channel. As you suggested, the cross section of ferritin is also displayed to clearly describe the nanopore structure (Supplementary Fig. 2).

The following sentence were added in the main text:

1) It is worth noting that, multi-level blockage signals are observed both in the detection of L-Cys and L-Hcy, which is considered as the simultaneous sensing for multiple molecules by the multi-channel ferritin sensor. The multi-level signal of L-Cys, as an example, is analyzed (Supplementary Fig. 9), in which the proportion of multi-level signals increases with the concentration of L-Cys, demonstrating the multi-channel feature of ferritin sensor. (Please see Page 5 left column, 1st paragraph with yellow highlight)

2) The ferritin nanopore sensing apparatus is prepared by inserting one ferritin molecule into the planar lipid bilayer, where four C4 channels are embedded in the lipid bilayer, and eight C3 channels are distributed on both sides of the membrane, with four on each side (Fig. 1c and Supplementary Fig. 2). (Please see Page 3 left column, 1st paragraph with yellow highlight)

3) The ionic current is believed to result from ions passing through only C3 channels rather than hydrophobic C4 channels, according to the reported ions transport pathway of ferritin. (Please see Page 3 right column, 1st paragraph with yellow highlight)

Supplementary Figure 2. Ferritin surface and cross-sectional diagram. All the eight C3 channels and six C4 channels are marked in blue and orange, respectively. Lipid bilayer is located on the XZ plane with one C4 channel and four C3 channels on each side.

2. *Clearly describe the origin of the ferritin sample at its first appearance. Was it ordered from a company or the team prepared it all by themselves? These information are important for others to replicate their results. Briefly introduce them in the maintext.*

Response 2: Thank you so much for your valuable suggestion. The ferritin is purchased from Sigma-Aldrich, which was previous described in Supplementary Information (now in main text). For more conveniently reading, we now mentioned the origin of ferritin in the maintext: “To be noted that the ferritin used in nanopore experiments is purchased from Sigma-Aldrich without further purification.” (Please see Page 2 right column, 1st paragraph with yellow highlight).

3. *MD simulation. It is appreciated to have MD simulation results. However, I don't quite get what kind of conclusion did you draw from the simulation? It seems that no valuable information was obtained from the simulation? Then why do we need the simulation at all?*

Response 3: Thanks for your comments. The coarse-grained molecular dynamics (CG-MD) simulation and multiscale (QM/AA/CG) MD simulation are used in this work. The long-time GC-MD simulation clearly demonstrated the structure of ferritin-lipid bilayer system, in which the four hydrophobic C4 channels are stably embedded in the lipid bilayer to construct the ferritin transmembrane nanopore. The multiscale MD simulation demonstrated the coordination model of Cu-Cys complex in C3 channel. The MD simulation provides the complementary understanding on how the ferritin stabilizes within the membrane and how the cysteine-based analyte is bound in the

C3 channel. Thus, we believe the MD simulation is helpful in both understanding the nanopore sensor constructing and cysteine-based analytes detection.

4. It is called a ferritin, does it naturally contain any bound Fe²⁺ or Fe³⁺ ion in the protein at all? The authors then used Cu²⁺ to realize biothiol sensing. Why use Cu²⁺ if this protein seems to be a Fe²⁺ or Fe³⁺ binding protein?

Response 4: Thank you so much for your valuable comments. Ferritin is an iron storage protein, with Fe³⁺ stored in the hollow cavity as iron oxide (*PNAS*, 2014, **111**, 7925-7930). As reported, Fe(III) can be reduced as Fe(II) and released from iron oxide core through C3 channel in the presence of reducing agents, such as flavin mononucleotide, and chelating agents like 2,2'-bipyridine (*Biochem. J.* 1974, **143**, 311-315). In our experimental condition, the Fe³⁺/Fe²⁺ is hard to be released from the iron oxide core of ferritin. To further exclude the influence of Fe³⁺/Fe²⁺, the nanopore experiments were carried out with additional Fe³⁺-cysteine mixture and Fe²⁺-cysteine mixture. No obvious current blockage events were observed in both experiments (**Supplementary Fig. 13**), demonstrating the existence of Fe³⁺ or Fe²⁺ has no background signals. In addition, other metal ions including Zn²⁺ and Ni²⁺ were also explored as connectors for sensing L-Cys, however, no blockage signals were observed (**Supplementary Fig. 13**). Moreover, Cu²⁺ was also reported to have strong interaction with Cys 126 residue on ferritin C3 channel (*PNAS*, 2014, **111**, 7925-7930), which enlightens us to select Cu²⁺ as the assistant sensing cysteine-based analytes.

The following sentence was added in the main text. “Besides, other metal ions including Fe³⁺, Fe²⁺, Zn²⁺, and Ni²⁺ were also explored as connectors for sensing L-Cys, however, no blockage signals were observed, implying the specificity of Cu²⁺ in this system (**Supplementary Fig. 13**).” (Please see **Page 5 right column, 1st paragraph with yellow highlight**).

Supplementary Figure 13. Current traces of (a) Cys-Fe³⁺, (b) Cys-Fe²⁺, (c) Cys-Zn²⁺, and (d) Cys-Ni²⁺ complexes detection. There is no obvious current blockage signals observed for all experiments. All the experiments are carried out in 1 M KCl buffer with Cys (80 μM) and Fe³⁺ (80 μM), Fe²⁺ (80 μM), Zn²⁺ (80 μM), or Ni²⁺ (80 μM) in trans chamber at +40 mV.

5. Though I am not 100% sure, it seems that there are multiple equivalent Cu²⁺ binding sites on the same ferritin nanopore. If so, why don't we observe simultaneous binding of multiple Cu²⁺ and Cys/Hcy binding events.

Response 5: Thanks for your valuable comments. As you said, the ferritin nanopore is indeed a multi-channel nanopore sensor. The multi-level signals were also observed both in L-Cys and L-Hcy experiments, which is considered as the simultaneous sensing for multiple molecules by the multi-channel ferritin sensor. The multi-level signals of L-Cys, as an example, were displayed (**Supplementary Fig. 9**). The following sentence was added in the main text. "It is worth noting that, multi-level blockage signals are observed both in the detection of L-Cys and L-Hcy, which is considered as the simultaneous sensing for multiple molecules by the multi-channel ferritin sensor. The multi-level signal of L-Cys, as an example, is analyzed (**Supplementary Fig. 9**), in which the

proportion of multi-level signals (level 2 and 3) increases with the concentration of L-Cys, demonstrating the multi-channel feature of ferritin sensor.” (Please see Page 5 left column, 1st paragraph with yellow highlight)

Supplementary Figure 9. Multi-level signals in Cys-Cu complexes detection. (a) Current trace of Cys-Cu complex detection. Typical multi-level signals are marked and displayed below. (b) The frequency of multi-level (level 2 and level 3) events increases with the L-Cys concentration. The experiments are carried out in 1 M KCl buffer with Cu²⁺ (80 μM) and Cys (20, 40, 60, and 80 μM) in trans chamber at +40 mV.

6. *Simultaneous discrimination of Hcy and Cys using the same pore. To acclaim nanopore discrimination of Hcy and Cys, the authors should provide simultaneous discrimination of Hcy and Cys using the same nanopore sensor and show a continuous trace lasting for a few mins. If this nanopore sensor does have the resolution to discriminate between Cys and Hcy, then two types of*

events should be alternatively observed in the same continuous trace. The corresponding event scatter plot results should also show two clear populations. Please note that this demonstration should not be skipped.

Response 6: Thanks for your valuable comments. The simultaneous detection for Cys and Hcy was carried out. As shown in **Supplementary Fig. 8**, the signals caused by Cys and Hcy are simultaneously observed in the same continuous trace, in which the Hcy displays larger current blockage. Two populations are also observed in scatter plot and histogram. However, the frequency of Hcy events is lower than that of Cys, although their concentrations are the same, which may be related to the stronger interaction of C3 channel with Cys-Cu complex than Hcy-Cu complex. The following sentence was added in the main text. “The Cys and Hcy events are simultaneously observed in the continuous recording, displaying the distinct discrimination (**Supplementary Fig. 8**).” (Please see **Page 5 left column, 1st paragraph with yellow highlight**)

Supplementary Figure 8. Simultaneous detection for Cys and Hcy. (a) Current trace of simultaneous detection for Cys (blue) and Hcy (green). The extracted current values are marked by red line. (b-c) Scatter plot and histogram for Cys (blue) and Hcy (green). The experiment is carried out in 1 M KCl buffer with Cu^{2+} (80 μM), Cys (40 μM) and Hcy (40 μM) in trans chamber at +40 mV.

7. Previous work demonstrating nanopore discrimination of Cys and Hcy. It seems that the authors fail to cite any previous reference of nanopore discrimination of Cys and Hcy (*Chem. Commun.*, 2019,55, 9311-9314; *Nat Commun* 10, 5668 (2019).) The authors also acclaimed that “the discrimination between L-Cys and L-Hcy has not been previously reported with non-engineered nanopore tools”. I guess no one cares whether a nanopore is engineered or not. People care whether this sensing task has been previously done, how it was done and how well it has been done. So it is more appropriate to cite relevant works, describe the concept difference between your idea and theirs. Compare their pros and cons.

Response 7: Thanks for your suggestion. The relevant references were cited and discussed in the manuscript. The following sentences were added in the main text.

Lu *et al.* successfully differentiated L-Cys and L-Hcy assisted by synthetic complex probe with mutant AeL nanopore.^[55] Cao *et al.* also achieved the differentiation between them by introducing [AuCl₄]⁻ as a bridge with methionine-mutated MspA nanopore, in which the sulfur atom on methionine plays a key role.^[56] With three reactive cysteine residues in each C3 channel, the ferritin nanopore is considered as a proper sensor for the discrimination between L-Cys and L-Hcy. (Please see **Page 4 right column, 1st paragraph with yellow highlight**)

[55] Lu, Y., Wu, X.-Y., Ying, Y.-L. & Long, Y.-T. Simultaneous single-molecule discrimination of cysteine and homocysteine with a protein nanopore. *Chem. Commun.* **55**, 9311 (2019).

[56] Cao, J. et al. Giant single molecule chemistry events observed from a tetrachloroaurate(III) embedded Mycobacterium smegmatis porin A nanopore. *Nat. Commun.* **10**, 5668 (2019).

8. As recently reported, a nanopore bound with Cu²⁺ ion is generally sensitive to all amino acids (*Nat Methods* 21, 609–618 (2024).). Another earlier study of Cu²⁺ nanopore for amino acid sensing was found as well (*Angew. Chem. Int. Ed.*, 51: 9606-9609). If that is the case, why do you observe binding of thiol containing amino acids? I am quite shocked that the *Angew Chemie* paper has not been cited at all. The *Nat Methods* paper was cited but not discussed and compared with this technique at all?

Response 8: Thanks for your valuable comments. The relevant references were cited and discussed in the manuscript. The following sentences were added in the main text.

It is worth noting that Cu^{2+} has previously been reported as connectors to help detect amino acids in nanopore methods, where amino acids are thought to bind to Cu^{2+} via their amino and carboxyl groups.^[9, 65] For example, Zhang *et al.* utilized MspA nanopore to detect 20 amino acids with the aid of Cu^{2+} .^[9] However, ferritin C3 channel is smaller than MspA nanopore, leading to a stronger electric field at entrance. Thus, only the form of most stable S-Cu-N rather than N-Cu-O strongly remains in the channel and generates signals. The sulfhydryl group on analytes is necessary for nanopore sensing with the ferritin nanopore. (Please see Page 5 right column, 2nd paragraph with yellow highlight).

[9] Zhang, M. et al. Real-time Detection of 20 Amino Acids and Discrimination of Pathologically Relevant Peptides with Functionalized Nanopore. *Nat. Methods* **21**, 609-618 (2024).

[65] Boersma, A.J. & Bayley, H. Continuous Stochastic Detection of Amino Acid Enantiomers with a Protein Nanopore. *Angew. Chem. Int. Ed.* **51**, 9606-9609 (2012).

9. *Quantification issue. Being a Cys/Hcy sensor, it seems that at a higher concentration of Cys/Hcy, the rate of event appearance is actually dropping. If that is the case, it means that this nanopore sensor has a big problem to deal with quantification of Cys/Hcy, which is quite disadvantageous.*

Response 9: Thanks for your valuable comments. The abovementioned situation only occurs when the Cys concentration is larger than the Cu^{2+} concentration. The event frequency is positively correlated with the Cys concentration when the Cys concentration is lower than the Cu^{2+} concentration. As we described in this manuscript (Page 5 right column, 2nd paragraph), the Cu^{2+} is able to binding one or two cysteine molecules to form different Cys-Cu complexes (Cys-Cu, Cu:Cys=1:1 and Cys-Cu-Cys, Cu:Cys=1:2), in which only the Cys-Cu complex can be detected in our system. When the Cys concentration is lower than Cu^{2+} , the event frequency increases with the Cys concentration. However, when the Cys concentration exceeds Cu^{2+} , the event frequency decreases as the Cys concentration continues to increase due to the partial transformation from Cys-Cu complex to Cys-Cu-Cys complex. To avoid the misunderstanding, the sentence “To be noted that the concentration of Cu^{2+} is fixed at 80 μM , while the concentrations of Cys are set at 40, 80, 120, and 160 μM , corresponding to the ratios of 1:0.5, 1:1, 1:1.5, and 1:2, respectively” was added in the main text. (Please see Page 5 right column, 2nd paragraph with yellow highlight)

To deal with the quantification issue, the qualification experiment for cysteine was carried out.

We first detected the cysteine samples in different concentration (20, 40, 60, 80 μM) with Cu^{2+} concentration fixed at 80 μM . The concentration standard curve was fitted according to the acquired event frequency. Then the actual sample (L-cysteine capsule) was dissolved and detected in three independent nanopore experiments. According to the fitted concentration standard curve, we acquired the calculated L-cysteine weight percentage as $16.6 \text{ wt}\% \pm 1.9 \text{ wt}\%$, which is close to the indicated value (18 wt%). The following sentences were added in the main text. “In addition, to test the qualification capacity of ferritin nanopore, the qualification experiments for L-Cys are carried out. The L-Cys concentration standard curve is fitted according to the acquired event frequency from standard solutions (Supplementary Fig. 10). Then, the actual sample (L-cysteine capsule) is dissolved and detected in three independent nanopore experiments. According to the fitted concentration standard curve, we acquired the calculated L-cysteine weight percentage as $16.6 \text{ wt}\% \pm 1.9 \text{ wt}\%$, close to the indicated value (18 wt%), revealing the well qualification capacity of ferritin nanopore (Supplementary Fig. 10).” (Please see Page 5 left column, 2nd paragraph with yellow highlight)

Supplementary Figure 10. Quantitative experiments for L-cysteine. (a) Event frequency of L-cysteine in different concentrations (20, 40, 60, 80 μM) with fixed 80 μM Cu^{2+} and the fitted concentration standard curve. (b) Comparison between calculated and indicated L-cysteine fraction in the L-cysteine capsule. The indicated weight percentage of L-cysteine is 18 wt% in the commercial L-cysteine capsule. In our experiment, 30 mg drug powder was dissolved in 10 mL H_2O and then 10 μL solution was pipetted into KCl buffer (previously mixed with Cu^{2+}) to prepare 1 mL test solution with 80 μM Cu^{2+} and unknown L-cysteine. Three independent nanopore

experiments were carried out and the statistical results of events frequency were acquired as 286 /5 min, 246 /5 min and 267 /5 min, respectively. According to the fitted formula in (a), the concentrations of L-cysteine were calculated and converted into weight percentage as 18.5 wt%, 14.7 wt%, and 16.7 wt%, respectively. Thus, the calculated weight percentage of L-cysteine was acquired as $16.6 \text{ wt}\% \pm 1.9 \text{ wt}\%$, close to the indicated value (18 wt%).

10. Fig 5, it seems that the discrimination between CK and CR seems to be not achieved? Does that mean the nanopore can only discriminate between CF and CK or CR? If that is true, it means that the nanopore resolution seems to be disadvantageous. I suggest removing the event populations of CK to SI. Leave only events that are clearly discriminated in the main text figure.

Response 10: Thanks for your comments. In this revision, we demonstrate the discrimination between CK and CR under conditions of higher salt concentration. As described in previous works, the charge, polarity and size of analytes influence the $\Delta I/I_0$ (*Nat Methods*, 2024, **21**, 92–101; *Nat Methods*, 2024, **21**, 102–109; *Nat Methods*, 2024, **21**, 609–618). The electrostatic interaction is believed to dominate in such a strongly negatively charged channel (**Supplementary Fig. 20**), making it difficult to distinguish between the similarly positively charged CK and CR despite their difference in volume. The ionic strength was improved by using higher concentration KCl buffer (2 M KCl) in nanopore experiments to decrease the electrostatic interaction between CR/CK and C3 channel. In this condition, the blockage fraction is dominated by volume exclusion effect, which leads to a larger $\Delta I/I_0$ for larger CR (**Supplementary Fig. 21**).

The following sentences were added in the maintext. “As reported, all the charge, polarity and size of analytes can influence the current blockage fraction in nanopore analysis.^[6, 9, 10] In our experiments, the blockage fractions of CK, CR, and CF are all smaller than that of L-Cys, with CF having the smallest $\Delta I/I_0$, and the difference between CK and CR is minimal (**Figs. 5e, f**). It is understood as that the smaller L-Cys is able to enter into the deeper site of the channel causing larger blockage fraction, while the dipeptides have too large volumes to enter deeper, causing smaller blockage fractions. The CF has strongest hydrophobicity and largest volume in all of them displaying the smallest blockage fraction. Given that lysine (K) and arginine (R) are hydrophilic polar amino acids, CK and CR are more likely to penetrate deeper into the C3 channel compared to CF, resulting in larger blockage fractions than CF. The charge interaction is believed to dominate in

such a strongly negatively charged C3 channel, making it difficult to distinguish between the similarly positively charged CK and CR despite their difference in volume. To decrease the charge interaction between CR/CK and C3 channel, we improve ionic strength by using higher concentration KCl buffer (2 M KCl) in nanopore experiments (Supplementary Fig. 21). In this condition, the blockage fraction is dominated by volume exclusion effect, which leads to a larger $\Delta I/I_0$ for larger CR.” (Please see Page 8 left column, 1st paragraph with yellow highlight)

To make readers more intuitively know what analytes are detected in our work, the CK events are considered to remain in main text.

[6] Zhang, Y. et al. Peptide Sequencing Based on Host–guest Interaction-assisted Nanopore Sensing. *Nat. Methods* **21**, 102-109 (2024).

[9] Zhang, M. et al. Real-time Detection of 20 Amino Acids and Discrimination of Pathologically Relevant Peptides with Functionalized Nanopore. *Nat. Methods* **21**, 609-618 (2024).

[10] Wang, K. et al. Unambiguous Discrimination of All 20 Proteinogenic Amino Acids and Their Modifications by Nanopore. *Nat. Methods* **21**, 92-101 (2024).

Supplementary Figure 20. Surface charge distribution map of the C3 channel. The whole C3 channel is negatively charged.

Supplementary Figure 21. Discrimination between CR and CK. (a) Current traces of CR-Cu²⁺ and CK-Cu²⁺ complexes detection. (b) Histogram for CR and CK. All the experiments are carried out in 2 M KCl buffer (pH 7.4) with CR (80 μM) or CK (80 μM) and Cu²⁺ (80 μM) in trans chamber at +40 mV.

11. Explain why CD and CK cause no signal at all?

Response 11: Thanks for your suggestions. Both CD and CE are negatively charged in our experimental condition. To explore if the direction of electrophoretic force hinders the generation of blockage events for negatively charged CD and CE, the testing solution with Cu²⁺ (80 μM) and CE (80 μM), as an example, was added in cis side to be detected with ferritin nanopore at +40 mV. No obvious current blockage events were observed in this situation, indicating the translocation direction of CE is not the reason why it doesn't cause blockage signals. (Fig. R1)

Yang *et al.* reported that positively charged and neutral molecules can enter the interior of ferritin, while negatively charged molecules have difficulty in entering (*Biophys. J.* 1996, **71**, 1587-1595). As shown in **Supplementary Fig. 20**, the C3 channel is negatively charged, making it difficult for negatively charged analytes to enter the channel due to electrostatic repulsion. In our experiments, the uncharged CF and positively charged CR and CK can cause current blockage signals, while negatively charged CD and CE cause no signals. Therefore, we believe the electrostatic repulsion is the reason why CD and CE do not generate a signal.

The following sentences were added in the main text. “Negatively charged molecules, as reported by Yang *et al.*, are hard to enter the cavity of ferritin via negatively charged C3 channel due to electrostatic repulsion,^[71] which can explain why the negatively charged CD and CE causes no signals.” (Please see **Page 7 right column, 2nd paragraph with yellow highlight**)

[71] Yang, X. & Chasteen, N.D. Molecular diffusion into horse spleen ferritin: a nitroxide radical spin probe study. *Biophys. J.* **71**, 1587-1595 (1996).

Figure R1. Current traces of CE-Cu²⁺ complex detection. The experiment is carried out in 1 M KCl buffer with Cu²⁺ (80 μM) and CE (80 μM) in cis chamber at +40 mV.

Supplementary Figure 20. Surface charge distribution map of the C3 channel. The whole C3 channel is negatively charged.

12. The pore is noisy when a high voltage is applied, meaning that it can only be used at a low applied bias, such as 40 mV. While most biological nanopores can withstand 200 mV bias without a problem. This means that this ferritin pore is disadvantageous in the resolution.

Response 12: Thanks for your comments. To test the detection capability of ferritin nanopore at a high voltage, we performed the measurement of Cys at 80 mV, a commonly used voltage in nanopore analysis (*Nano Lett.* 2024, **24**, 14118–14124; *Adv. Mater.* 2023, **35**, 2300589), in which the characteristic signals could also be detected. (**Supplementary Fig. 15**)

Supplementary Figure 15. Nanopore detection for Cys-Cu²⁺ complex at 80 mV. (a-b) Current trace and scatter plot of Cys-Cu²⁺ complex detection. The characteristic events for Cys-Cu²⁺ are marked in red. Plots on the left with broad $\Delta I/I_0$ distribution are background signals. (c) Background signals of nanopore recording without any analytes at 80 mV.

13. Eventually, based on all above discussions, though I found this newly discovered ferritin nanopore quite interesting and novel, it is generally disadvantageous as a nanopore sensor by having a low resolution, an unclear structure and binding site for future engineering. Plenty of grammar and typo issues exist as well. The authors also failed to cite strongly relevant previous works with similar concepts. It is always good to new types of nanopores, good for the field. However, to qualify publication at a high impact journal like Nature Communcations, this new nanopore should show its unique sensing properties and advantages. I hope my above suggestions may help the authors to further improve this manuscript.

Response 13: Thanks for your time, effort and valuable comments. The relevant previous works were cited and the grammar was also double checked and corrected. Thanks again for your professional suggestions which help us significantly improve this manuscript.

Reviewer: 2

Comments: Nanopore sensing, renowned for its label-free, highly sensitive, accurate, cost-effective, high-throughput, and portable characteristics, has garnered significant attention in recent years. A key area of focus is the development of new nanopores with enhanced stability and spatial resolution. Through a combination of experimental results and molecular dynamics (MD) simulations, the author provides compelling evidence that a single ferritin molecule can be inserted into a lipid bilayer, forming a stable ionic channel. Further exploration demonstrated the sensing capabilities for cysteine, homocysteine, and cysteine-containing dipeptides. High-quality experimental data and computational simulations elucidated the mechanism by which the ferritin nanopore detects these molecules, facilitated by Cu^{2+} . This study underscores the potential of ferritin for ultrasensitive single-molecule sensing, given its pore size of 3-5 Å. However, the ability to identify dipeptides has not been fully explored, and the validation of some hypotheses remains insufficient. Further revisions are needed to clarify the methodology and its robustness.

Response: Thank you so much for your time, effort, and expertise in reviewing our work. Your mentioned questions have been responded as follows.

1. It is somewhat counterintuitive that a spherical protein could be stabilized within a lipid bilayer. Is this an isolated example, or is it possible to find more spherical proteins that can be inserted into lipid bilayers?

Response 1: Thanks for your questions. To the best of our knowledge, this is the first report of a globular protein inserted into a lipid bilayer. Repetitive insertion experiments and molecular dynamics simulation support that ferritin can insert into and stably exist within the lipid bilayer. We speculate that the uniformly distributed hydrophobic C4 regions provide favorable binding sites for lipid bilayer, allowing ferritin to stably embed within the membrane. Wen *et al.* also reported that by modifying hydrophobic cholesterol targets, the globular DNA origami structure can more stably remain on lipid-covered SiN_x nanopore (*Nano Lett.* 2023, **23**, 788–794), which revealing the critical role of hydrophobic targets in stabilizing spherical molecules. Therefore, we believe it is possible to find more spherical proteins that can be inserted into lipid bilayer if they have suitable hydrophobic sites.

2. The scatter plot of CK and CR nearly overlaps. Can these two dipeptides be differentiated under altered experimental conditions?

Response 2: Thanks for your valuable comments. In this revision, we demonstrate the discrimination between CK and CR under conditions of higher salt concentration. As described in previous works, the charge, polarity and size of analytes influence the $\Delta I/I_0$ (*Nat Methods*, 2024, **21**, 92–101; *Nat Methods*, 2024, **21**, 102–109; *Nat Methods*, 2024, **21**, 609–618). The electrostatic interaction is believed to dominate in such a strongly negatively charged channel (**Supplementary Fig. 20**), making it difficult to distinguish between the similarly positively charged CK and CR despite their difference in volume. The ionic strength was improved by using higher concentration KCl buffer (2 M KCl) in nanopore experiments to decrease the electrostatic interaction between CR/CK and C3 channel. In this condition, the blockage fraction is dominated by volume exclusion effect, which leads to a larger $\Delta I/I_0$ for larger CR (**Supplementary Fig. 21**).

The following sentences were added in the maintext. “As reported, all the charge, polarity and size of analytes can influence the current blockage fraction in nanopore analysis.^[6, 9, 10] In our experiments, the blockage fractions of CK, CR, and CF are all smaller than that of L-Cys, with CF having the smallest $\Delta I/I_0$, and the difference between CK and CR is minimal (**Figs. 5e, f**). It is understood as that the smaller L-Cys is able to enter into the deeper site of the channel causing larger blockage fraction, while the dipeptides have too large volumes to enter deeper, causing smaller blockage fractions. The CF has strongest hydrophobicity and largest volume in all of them displaying the smallest blockage fraction. Given that lysine (K) and arginine (R) are hydrophilic polar amino acids, CK and CR are more likely to penetrate deeper into the C3 channel compared to CF, resulting in larger blockage fractions than CF. The charge interaction is believed to dominate in such a strongly negatively charged C3 channel, making it difficult to distinguish between the similarly positively charged CK and CR despite their difference in volume. To decrease the charge interaction between CR/CK and C3 channel, we improve ionic strength by using higher concentration KCl buffer (2 M KCl) in nanopore experiments (**Supplementary Fig. 21**). In this condition, the blockage fraction is dominated by volume exclusion effect, which leads to a larger $\Delta I/I_0$ for larger CR.” (Please see **Page 8 left column, 1st paragraph with yellow highlight**)

Supplementary Figure 20. Surface charge distribution map of the C3 channel. The whole C3 channel is negatively charged.

Supplementary Figure 21. Discrimination between CR and CK. (a) Current traces of CR-Cu²⁺ and CK-Cu²⁺ complexes detection. (b) Histogram for CR and CK. All the experiments are carried out in 2 M KCl buffer (pH 7.4) with CR (80 μM) or CK (80 μM) and Cu²⁺ (80 μM) in trans chamber at +40 mV.

3. The underlying mechanism behind the absence of signals from CD and CE should be further investigated. There is insufficient evidence to support the conjecture that the presence of COOH groups may prevent peptides from binding to nanopores. The negative charge may affect the translocation of the dipeptide. Additional supporting experiments or simulations should be

conducted.

Response 3: Thank you for pointing it out. To check our previous speculation that extra COOH group prevent Cu-dipeptide complex from binding to C3 channel, we tried to mix glutaric acid with Cys-Cu testing solution. However, we still observed large amount of blockage events caused by Cys-Cu complex, revealing that extra COOH groups cannot prevent Cys-Cu complex from binding to the C3 channel. We now consider it seems not a reasonable explanation that the presence of COOH groups may prevent Cu-dipeptide complex from binding to nanopores.

As you said, the negative charge may affect the translocation of the dipeptide. To explore if the direction of electrophoretic force hinders the generation of blockage events for negatively charged CD and CE, the testing solution with Cu^{2+} (80 μM) and CE (80 μM), as an example, was added in cis side to be detected with ferritin nanopore at +40 mV. No obvious current blockage events were observed in this situation, indicating the translocation direction of CE is also not the reason why it doesn't cause blockage signals. (Fig. R1)

Yang *et al.* reported that positively charged and neutral molecules can enter the interior of ferritin, while negatively charged molecules have difficulty in entering (*Biophys. J.* 1996, **71**, 1587-1595). As shown in **Supplementary Fig. 20**, the C3 channel is negatively charged, making it difficult for negatively charged analytes to enter the channel due to electrostatic repulsion. In our experiments, the uncharged CF and positively charged CR and CK can cause current blockage signals, while negatively charged CD and CE cause no signals. Therefore, we believe the electrostatic repulsion is the reason why CD and CE do not generate a signal.

The following sentences were added in the maintext. “Negatively charged molecules, as reported by Yang *et al.*, are hard to enter the cavity of ferritin via negatively charged C3 channel due to electrostatic repulsion,^[71] which can explain why the negatively charged CD and CE causes no signals.” (Please see **Page 7 right column, 2nd paragraph with yellow highlight**)

[71] Yang, X. & Chasteen, N.D. Molecular diffusion into horse spleen ferritin: a nitroxide radical spin probe study. *Biophys. J.* **71**, 1587-1595 (1996).

Figure R1. Current traces of CE-Cu²⁺ complex detection. The experiment is carried out in 1 M KCl buffer with Cu²⁺ (80 μM) and CE (80 μM) in cis chamber at +40 mV.

Supplementary Figure 20. Surface charge distribution map of the C3 channel. The whole C3 channel is negatively charged.

4. The author claimed that “the smaller L-Cys is able to enter into the deeper site of the channel causing a larger blockage fraction.” However, this claim is contradicted by the observation that R, which has a larger volume than K, shows a deeper blockage in the CR dipeptide.

Response 4 : Thanks for your valuable comments. In the origin manuscript, only the volume was considered in understanding the blockage signals. In fact, all the charge, polarity and size of analytes can influence the $\Delta I/I_0$ (*Nat Methods*, 2024, **21**, 92–101; *Nat Methods*, 2024, **21**, 102–109; *Nat Methods*, 2024, **21**, 609–618). In our experiments, the blockage fractions of CK, CR, and CF are all smaller than that of L-Cys, with CF having the smallest $\Delta I/I_0$, and the difference between CK and CR is minimal. It is understood as that the smaller L-Cys is able to enter into the deeper site of the channel causing larger blockage fraction, while the dipeptides have too large volumes to enter deeper, causing smaller blockage fractions. The CF has strongest hydrophobicity and largest volume

in all of them displaying the smallest blockage fraction. Given that lysine (K) and arginine (R) are hydrophilic polar amino acids, CK and CR are more likely to penetrate deeper into the C3 channel compared to CF, resulting in larger blockage fractions than CF.

The charge interaction is believed to dominate in such a strongly negatively charged C3 channel (Supplementary Fig. 20), making it difficult to distinguish between the similarly positively charged CK and CR despite their difference in volume. To decrease the charge interaction between CR/CK and C3 channel, we improve ionic strength by using higher concentration KCl buffer (2 M KCl) in nanopore experiments. In this condition, the blockage fraction is dominated by volume exclusion effect, which leads to a larger $\Delta I/I_0$ for larger CR. (Supplementary Fig. 21)

The following sentences were added in the main text. “As reported, all the charge, polarity and size of analytes can influence the current blockage fraction in nanopore analysis.^[6, 9, 10] In our experiments, the blockage fractions of CK, CR, and CF are all smaller than that of L-Cys, with CF having the smallest $\Delta I/I_0$, and the difference between CK and CR is minimal (Figs. 5e, f). It is understood as that the smaller L-Cys is able to enter into the deeper site of the channel causing larger blockage fraction, while the dipeptides have too large volumes to enter deeper, causing smaller blockage fractions. The CF has strongest hydrophobicity and largest volume in all of them displaying the smallest blockage fraction. Given that lysine (K) and arginine (R) are hydrophilic polar amino acids, CK and CR are more likely to penetrate deeper into the C3 channel compared to CF, resulting in larger blockage fractions than CF. The charge interaction is believed to dominate in such a strongly negatively charged C3 channel, making it difficult to distinguish between the similarly positively charged CK and CR despite their difference in volume. To decrease the charge interaction between CR/CK and C3 channel, we improve ionic strength by using higher concentration KCl buffer (2 M KCl) in nanopore experiments (Supplementary Fig. 21). In this condition, the blockage fraction is dominated by volume exclusion effect, which leads to a larger $\Delta I/I_0$ for larger CR.” (Please see Page 8 left column, 1st paragraph with yellow highlight)

[6] Zhang, Y. et al. Peptide Sequencing Based on Host–guest Interaction-assisted Nanopore Sensing. *Nat. Methods* **21**, 102-109 (2024).

[9] Zhang, M. et al. Real-time Detection of 20 Amino Acids and Discrimination of Pathologically Relevant Peptides with Functionalized Nanopore. *Nat. Methods* **21**, 609-618 (2024).

[10] Wang, K. et al. Unambiguous Discrimination of All 20 Proteinogenic Amino Acids and Their

Modifications by Nanopore. *Nat. Methods* **21**, 92-101 (2024).

Supplementary Figure 20. Surface charge distribution map of the C3 channel. The whole C3 channel is negatively charged.

Supplementary Figure 21. Discrimination between CR and CK. (a) Current traces of CR-Cu²⁺ and CK-Cu²⁺ complexes detection. (b) Histogram for CR and CK. All the experiments are carried out in 2 M KCl buffer (pH 7.4) with CR (80 μM) or CK (80 μM) and Cu²⁺ (80 μM) in trans chamber at +40 mV.

Reviewer: 3

Comments: In their manuscript, Yun-Dong et al. describe the application of horse spleen ferritin (a mostly L-ferritin oligomer) as a nano-pore bio-sensing analytical device. The authors use single-channel voltage clamp recordings to demonstrate that ferritin can function as a stable trans-membrane ion channel. The authors combine their experimental work with computational simulations to provide insights into the molecular mechanism of transport through the ferritin channels. The authors test the ferritin nano-pore for the detection of amino acids, using the binding of copper ions to enhance the sensing capabilities. They then use an elaborate analysis of the system by adding/changing different parts of the system (e.g., Cu ions, buffers) to elucidate the molecular basis of the sensing.

Overall, this research is innovative, with compelling results and discussion; thus, I strongly recommend its publication. However, there are several weaknesses/shortcomings that I recommend addressing before publication:

Response: Thank you so much for your time, effort, and expertise in reviewing our work. Thanks for your recognition of our work. Your mentioned questions have been responded as follows.

1. The treatment of the protein sample is too superficial. Horse spleen ferritin is mostly L-ferritin but not only L-ferritin. Thus, the authors should at least characterize the sample used (e.g., SDS and Native PAGE, size-exclusion chromatography, etc.) to determine the actual composition. Preferably, they should repeat at least one experiment with a recombinantly expressed and purified protein (L-ferritin is easily purified).

Response 1: Thanks for your valuable comments. The SDS- and Native-PAGE were performed to characterize ferritin sample. As we all know, the L-chain of horse spleen ferritin is 19 kDa and the H-chain is 21 kDa (*Protein Science*, 2023, **32**, e4543.). In the SDS-PAGE experiment, only a single band below 20 kDa was observed, with no bands above 20 kDa detected, indicating that the ferritin sample predominantly consists of L-chains. (**Supplementary Fig. 1a**) In the Native-PAGE experiment, the band corresponding to 440 kDa indicates that ferritin sample is a complete 24-mer without cleavage. (**Supplementary Fig. 1b**)

Supplementary Figure 1. SDS- and Native-PAGE for ferritin sample. (a) SDS-PAGE analysis of ferritin sample. (b) Native-PAGE analysis of ferritin sample. M: marker, Ferritin: ferritin sample. The SDS-PAGE is carried out with 12% gel at 120 V. The Native-PAGE is carried out with 15% gel at 150 V. In the SDS-PAGE experiment, only a single band below 20 kDa was observed, with no bands above 20 kDa detected, indicating that the ferritin sample predominantly consists of L-chains.

2. *If the authors' claims are correct, the system has four pores above and below the membrane. In such a case, one can calculate the number of events where more than one pore is blocked simultaneously and how the propensity of these events changes with the concentration of analytes. Can you find that in your data?*

Response 2: Thanks for your valuable comments. As you said, we observed the multi-level signals both in L-Cys and L-Hcy experiments, which is considered as the simultaneous sensing for multiple molecules by the multi-channel ferritin sensor. We studied how the frequency of multi-level signals changes with the concentration of L-Cys. As a result, the proportion of multi-level signals increases with the concentration of L-Cys. (**Supplementary Fig. 9**) The following sentence was added in the main text. “It is worth noting that, multi-level blockage signals are observed both in the detection of L-Cys and L-Hcy, which is considered as the simultaneous sensing for multiple molecules by the multi-channel ferritin sensor. The multi-level signal of L-Cys, as an example, is analyzed (**Supplementary Fig. 9**), in which the proportion of multi-level signals (level 2 and 3) increases with the concentration of L-Cys, demonstrating the multi-channel feature of ferritin sensor.” (Please see **Page 5 left column, 1st paragraph with yellow highlight**)

Supplementary Figure 9. Multi-level signals in Cys-Cu complexes detection. (a) Current trace of Cys-Cu complex detection. Typical multi-level signals are marked and displayed below. (b) The frequency of multi-level (level 2 and level 3) events increases with the L-Cys concentration. The experiments are carried out in 1 M KCl buffer with Cu^{2+} (80 μM) and Cys (20 40 60 and 80 μM) in trans chamber at +40 mV.

3. *The role of three amino acids in the pore is discussed in detail. I recommend testing these claims by repeating at least one experiment with a mutant of L-ferritin replacing one (or more) of these amino acids.*

Response 3: Thanks for your valuable comments. In our manuscript, how the Cys-Cu complex bounded in C3 channel was explored by MD simulation, in which the His114, Cys126 and Glu130 play key roles in stabilizing the Cys-Cu complex in C3 channel. It is hard for us to prepare mutant ferritin on time due to our inexperience in bioengineering. An alternative solution was used here to solve this question.

The Zn^{2+} , as reported, has stronger interaction with these binding sites than Cu^{2+} (*PNAS*, 2014, **111**, 7925-7930). Thus, we introduced excess Zn^{2+} in the test solution to occupy these binding sites in C3 channel and analyzed how the Cys- Cu^{2+} events frequency becomes before and after introducing Zn^{2+} . As a result, an obviously lower events frequency was observed in the detection of Cys- Cu^{2+} complex in the presence of excess Zn^{2+} , accordingly revealing the key roles of these residues in binding Cys- Cu^{2+} complex. (Supplementary Fig. 18) To exclude the interference from potential Cys- Zn^{2+} complex, the background experiment was also carried out and no obvious current blockage signals were observed. (Figure R2.)

Following sentence was added in the main text. “To further verify the key roles of these residues, we introduce excess Zn^{2+} , which has stronger interaction than Cu^{2+} with these residues,^[34] in the detection for Cys-Cu complex to occupy these binding sites and analyze how the Cys-Cu events frequency becomes before and after introducing Zn^{2+} . As a result, an obviously lower events frequency is observed in the detection of Cys-Cu complex in the presence of excess Zn^{2+} , accordingly revealing the key roles of these residues in binding Cys-Cu complex. (Supplementary Fig. 18)” (Please see Page 7 right column, 1st paragraph with yellow highlight)

[34] Behera, R.K. & Theil, E.C. Moving Fe^{2+} from Ferritin Ion Channels to Catalytic OH Centers Depends on Conserved Protein Cage Carboxylates. *Proc. Natl. Acad. Sci. U.S.A.* **111**, 7925-7930 (2014).

Supplementary Figure 18. Events frequency in Cys- Cu^{2+} complex detection without (gray) and with (blue) Zn^{2+} . All the experiments are carried out in 1 M KCl buffer with 80 μM Cu^{2+} , 80 μM Cys and 240 μM Zn^{2+} (if added) in trans chamber at +40 mV.

Figure R2. Current trace of potential Cys-Zn²⁺ complex detection. There is no obvious current blockage signals observed. The experiment is carried out in 1 M KCl buffer with Zn²⁺ (80 μM) and Cys (80 μM) in trans chamber at +40 mV.

4. *While the paper is easily read and well organized, some annoying grammatical errors (e.g., problems with plural/singular messing "the" etc.) should be addressed. The term "proven" is used too lightly and should be avoided.*

Response 4: Thanks for your suggestions. The grammar was double checked and corrected. The term “proven” was replaced by “demonstrated” and “reported” in our revision.

1) The Cu²⁺ is **demonstrated** to bridge analytes and residues (Cys126 and Glu130) inside the ferritin C3 channel, leading to the blockage current signals. (Please see **Page 2 left column, 1st paragraph with yellow highlight**)

2) In detail, the Cys126 residue near the external entrance of C3 channel is **reported** to play a key role in absorbing Fe²⁺. (Please see **Page 2 right column, 1st paragraph with yellow highlight**)

This completes our reply. We thank the Referees for their valuable comments/suggestions and critical reading of our work. We hope that the changes we made will prove satisfactory. At the same time, we remain open to further suggestions as to how to improve the manuscript.

With warm regards,

Zhi-Yuan Gu & Co-Authors

Response to Reviewers' Comments

Reviewer: 1

Comments: The authors have successfully addressed all my review comments. The manuscript is now ready to be published.

Response: Thank you so much for your time, effort, and expertise in reviewing our work. Your constructive suggestions greatly improved the quality of our manuscript.

Reviewer: 2

Comments: The authors have addressed all the concerns raised during the first-round reviewing. In general, the work is carefully executed and the results are quite interesting. It may prompt researchers in the field to test many other proteins like ferritin as nanopores.

Response: Thank you so much for your time, effort, and expertise in reviewing our work. Your constructive suggestions greatly improved the quality of our manuscript.

Reviewer: 4

Comments: I have significant concerns regarding the use of horse spleen ferritin (purchased from Sigma-Aldrich) in this study, as it introduces multiple uncertainties that could compromise the validity and reproducibility of the results. First, the ferritin used appears to be in its holo-form (iron-loaded), given the information provided in the authors' response. Horse spleen ferritin can contain thousands of iron atoms, which could significantly influence the nanopore behavior and signal responses, particularly in experiments intended to study specific interactions with other metals like copper. The observed lack of blockage signals could be attributable to the overwhelming presence of iron rather than the intrinsic properties of the ferritin nanopore system. Furthermore, the choice of horse spleen ferritin without further purification raises additional issues of purity and consistency. Based on prior experience, commercially available horse spleen ferritin from Sigma-Aldrich often lacks sufficient purity for rigorous scientific studies and can yield inconsistent data. This raises questions about why the authors did not opt for higher-quality alternatives, such as human recombinant ferritin, which would offer better control over subunit composition (H or L), iron loading, and overall reproducibility. Thus, I can fully understand the concern raised by Reviewer 3. My observations and feedback on the authors' responses to Reviewer 3 are outlined below:

Response: Thank you so much for your time, effort, and expertise in reviewing our work. Your comments and questions have been properly addressed and responded.

First, to explore the effect of the iron core on the sensing property of ferritin, we have added additional Cys-Cu detecting experiments using horse spleen apo-ferritin (iron-unloaded). The ionic conductivity and sensing properties of apo-ferritin nanopore are identical to those of ferritin nanopore, which indicates that the iron core inside ferritin plays minor role in this sensing experiment. (Please see pages 10-12)

Second, the comprehensive characterizations, such as size-exclusion chromatography (SEC), SDS-PAGE and Native-PAGE, DLS analysis, and HRTEM for horse spleen ferritin sample have been now added in this revision. (Please see pages 5-7) Regarding the purity you questioned in the comments, we have two interpretations. The first is whether there are any contaminant proteins in the ferritin sample, and the second is whether the ferritin sample consists entirely of L-subunits. Based on our characterization, the ferritin sample used in this work does not significantly contain

contaminant proteins. As for the native horse spleen ferritin sample, we believe it is not pure entire L-chain ferritin but rather L-rich ferritin as reported in previous references. In the previous version, we described our ferritin sample as pure horse spleen L-ferritin because our SDS-PAGE experiment showed only a single band between 15 and 20 kDa corresponding to L-chains. However, some literatures indicated that native horse spleen ferritin is L-rich ferritin ($L > 90\%$) rather than pure L-ferritin, and the band corresponding to H-chains is difficult to detect by SDS-PAGE due to its low abundance. Therefore, to avoid misunderstanding, we have corrected "L-ferritin" to "L-rich ferritin" as most references suggested. We fully understand the reviewer's concern about the experimental reproducibility due to the uncertain H/L composition of the native horse spleen ferritin. To verify the experimental reproducibility, we have produced membrane-embedding testing, I-V testing, and Cys-Cu sensing with different batches of horse spleen ferritin (holo and apo-form), and the consistent experimental results were observed. Thus, we could conclude that our experiments with horse spleen ferritin are reproducible.

Third, the recombinantly expressed and purified human L-ferritin (LFn) has now been added to construct the LFn nanopore for Cys-Cu detection. LFn is also able to detect Cys-Cu but with the fluctuating current baseline. The difference in current baseline and Cys-Cu signals between human LFn and horse ferritin were also described and the possible reasons were accordingly discussed. The recombinantly expressed and purified human H-ferritin (HF_n) are also added to construct HF_n nanopore many times but never succeed. No membrane-embedding phenomenon is observed for all HF_n testing. Regarding heteropolymer ferritins with varied H/L ratios, we really appreciate your insightful suggestion. However, due to the low membrane-embedding ability of HF_n and the high fluctuating current baseline of LFn, it is neither practical nor necessary to perform these series of control experiments. Furthermore, the construction of heteropolymer ferritins-based nanopore is beyond the scope of the current work. (Please see pages 8-10)

Fourth, a quantitative model is built to describe the multi-level signals. The experimental probabilities of multi-level signals including state 0, state 1, state 2, state 3, and state 4 at five different Cys-Cu concentrations from 0 to 80 μM are calculated and summarized. Then, a binomial distribution model $\text{Bin}(4,q)$ is established to plot the predicted probability curves of the above five states. The experimental probabilities of multi-level signals are found to have a good correspondence with the predicted curves from the binomial distribution model. Through which, we

obtain the each-channel-blocked probability (q value) for each Cys-Cu concentration. After that, the relationship between each-channel-blocked probability (q value) and Cys-Cu concentrations was plotted and fitted quite well with the logistic model. Thus, the probability of each multi-level signal at a given concentration could be easily calculated by combining the logistic model and binomial distribution model. (Please see pages 13-16)

Finally, the key role of three residues (His114, Cys126, and Glu130) in sensing Cys-Cu was further explored through crystal structure, MD simulation and the concentration control experiment with Zn^{2+} . Thus, we verified the coordination style of Zn^{2+} in C3 channel and the competitive occupation effect of Zn^{2+} with Cys-Cu on the same binding sites. (Please see pages 17-22)

Your questions have been detailed responded as follows.

1. The authors have not adequately addressed the first question raised by Reviewer 3. Instead, they appear to have taken a simplistic approach by presenting gel data (SDS and PAGE) without providing comprehensive characterization of the horse spleen ferritin composition, as requested. Additionally, the authors did not repeat the experiments using recombinantly expressed and purified ferritin proteins, such as homopolymer L-ferritin or H-ferritin, which are widely utilized in the field. It remains unclear whether the reported nanopore sensing capability attributed to ferritin is also observed with homopolymer H-ferritin or heteropolymer ferritins composed of varying combinations of H and L subunits. Additionally, the effect of a large iron core present in horse spleen ferritin on the sensing properties of ferritin has not been explored and been largely ignored in these experiments.

Response 1: Thanks for your valuable comments. First, we have added comprehensive characterizations for the horse spleen ferritin sample, including size-exclusion chromatography (SEC), SDS-PAGE and Native-PAGE, DLS analysis, and HRTEM images (Supplementary Fig. 1). In SEC experiment, the main peak at 15.5 minutes represents the monomer ferritin, and the small peaks at 12 and 13 minutes represent the ferritin aggregates.^[1] The minor peak marked by * may be caused by salts, which was also observed in previous works.^[2-3] There are no other peaks, indicating the good purity of the ferritin sample. It is well accepted that the L-chain of horse spleen ferritin is ~19 kDa and the H-chain is ~21 kDa. As reported, the horse spleen ferritin primarily consists of L-chains (~90%).^[4-5] In our SDS-PAGE experiment, only a single band between 15 and 20 kDa was clearly observed, with no obvious bands above 20 kDa detected, indicating that the ferritin sample

predominantly consists of L-chains, which is consistent with previously works.^[5] In Native-PAGE experiment, the band corresponding to ~440 kDa indicates that ferritin sample is a intact L-rich ferritin without cleavage. The band corresponding higher molecular weight (it doesn't mean ~669 kDa but have not been totally expended) means the ferritin aggregates,^[6] which is consistent with the SEC experiments. The DLS analysis revealed that the ferritin sample has a diameter of ~11 nm which is the typical size of 24-mer ferritin.^[7] The HRTEM experiment also verify the intact spherical structure of ferritin sample. All above characterizations reveal that the ferritin sample used in this work is pure and intact L-rich ferritin. The following sentence was added in the main text. "To be noted that the ferritin sample used in nanopore experiments (Sigma-Aldrich) is comprehensively characterized by SEC, SDS-PAGE, Native-PAGE, DLS, and HRTEM experiments and identified as intact L-rich ferritin with spherical structure (Supplementary Fig. 1). For convenience, the "ferritin" was used to represent the L-rich horse spleen ferritin in this work." (Please see Page 2 right column, 1st paragraph with yellow highlight).

Supplementary Figure 1. Characterization for horse spleen ferritin sample. (a) SEC analysis for ferritin. The main peak at 15.5 minutes represents the monomer ferritin, and the small peaks at 12

and 13 minutes respectively represent the ferritin aggregates, which has been reported previously.^[1] The minor peak marked by * may be caused by salts, which was also observed in previous works.^[2-3] There are no other peaks, indicating good purity of the ferritin sample. (b) SDS- and Native-PAGE analysis for ferritin. M: marker, Ferritin: ferritin sample. The SDS-PAGE and Native-PAGE are carried out with 12% gel at 120 V and with 15% gel at 150 V, respectively. In the SDS-PAGE experiment, only a single band between 15kDa and 20 kDa was observed, with no obvious bands above 20 kDa detected, indicating that the ferritin sample predominantly consists of L-chains which is consistent with previously works.^[5] In Native-PAGE experiment, the intact 24-mer ferritin band is observed at ~440 kDa. The band corresponding higher molecular weight (it doesn't mean ~669 kDa but have not been totally expended) means the ferritin aggregates,^[6] which is consistent with the SEC experiments in (a). (c-d) DLS analysis and stained HTREM images for ferritin. Insertion is the size statistic for ferritin. The DLS and HRTEM experiments verify the intact spherical structure (~11 nm) of ferritin sample.^[7]

[1] Niitsu, Y. & Listowsky, I. Mechanisms for the formation of ferritin oligomers. *Biochemistry*. **12**, 4690-4695 (1973).

[2] Ladd Effio, C., Oelmeier, S.A. & Hubbuch, J. High-throughput characterization of virus-like particles by interlaced size-exclusion chromatography. *Vaccine* **34**, 1259-1267 (2016).

[3] Yamazaki, J. et al. Simultaneous quantification of oligo-nucleic acids and a ferritin nanocage by size-exclusion chromatography hyphenated to inductively coupled plasma mass spectrometry for developing drug delivery systems. *Anal. Methods* **14**, 2219-2226 (2022).

[4] Plath, L.D., Ozdemir, A., Aksenov, A.A. & Bier, M.E. Determination of Iron Content and Dispersity of Intact Ferritin by Superconducting Tunnel Junction Cryodetection Mass Spectrometry. *Anal. Chem.* **87**, 8985-8993 (2015).

[5] Srivastava, A.K., Reutovich, A.A., Hunter, N.J., Arosio, P. & Bou-Abdallah, F. Ferritin microheterogeneity, subunit composition, functional, and physiological implications. *Sci. Rep.* **13**, 19862 (2023).

[6] Zhang, S. et al. Modifications of Thermal-Induced Northern Pike (*Esox lucius*) Liver Ferritin on Structural and Self-Assembly Properties. *Foods* **11**, 2897 (2022).

[7] Song, N. et al. Ferritin: A Multifunctional Nanoplatfrom for Biological Detection, Imaging Diagnosis, and Drug Delivery. *Acc. Chem. Res.* **54**, 3313-3325 (2021).

Second, according to the reviewer's suggestion, we cooperated with Prof. Xi-Yun Yan and Prof. Ke-Long Fan at Institute of Biophysics, Chinese Academy of Sciences, the renowned group in human ferritin, to obtain recombinantly expressed and purified human L-ferritin (LFn) and human H-ferritin (HF_n), and repeated the Cys-Cu sensing experiment using recombinantly expressed and purified LFn nanopore. First, we experimentally constructed the LFn based nanopore sensor, recorded the I-V curve of LFn nanopore, and sensed the Cys-Cu complex with the LFn nanopore (Fig. R1). As results, the membrane-embedding ability of LFn was confirmed by a sharp increase during the current recording, and the I-V curve of LFn is similar to that of horse L-rich ferritin. Then, the Cys-Cu sensing experiment with LFn was carried out. Under +60 mV, the current baseline was observed keeping at ~25 pA and the Cys-Cu events (marked with red triangle) were observed (Fig. R1c). The Scatter plot and histogram of Cys-Cu events were drawn and analyzed (Fig. R1d, e). The $\Delta I/I_0$ of Cys-Cu events recorded with LFn nanopore is 0.338, which is close to that with horse ferritin as 0.331. The generation of Cys-Cu events was attributed to the residence of Cys-Cu in the C3 channel of LFn, which has the identical His114, Cys126, and Glu130 residues as horse ferritin (Fig. R2a). It is worthing noted that, although LFn is recombinantly expressed and purified sample, the LFn nanopore sensor displayed a more fluctuating current baseline and shorter dwell time of Cys-Cu events than that in horse ferritin testing. This was attributed to the different amino acid sequence in the C3 channel (Fig. R2), which is obtained from PDB files (horse spleen ferritin: 2w0o, LFn: 2ffx). For LFn, its Thr121 and Thr131 residues with longer side chain replace the Ala121 and Ser131 residues in horse ferritin, respectively, which is considered as the reason for more fluctuating ion current. As for the Cys-Cu events with shorter dwell time, it is speculated that the positively charged Arg120 residue on LFn repels the positively charged Cys-Cu and thus shortens its dwell time, while the neutral Gln120 on horse ferritin does not. For the HF_n, we have tried many times but didn't observe the membrane-embedding behavior. Regarding heteropolymer ferritins, we appreciate your insightful suggestion. However, the H-chains in ferritin is disadvantageous to its membrane-embedding ability according our experiments with HF_n, and the construction of heteropolymer ferritins-based nanopore is beyond the scope of the current work.

Figure R1. Nanopore experiments with LFn. (a) Typical current trace of single LFn inserting into lipid bilayer at +150 mV. (b) Comparison of the I-V curves of LFn (blue) and horse ferritin (black). (c) Current traces of Cys-Cu detection with LFn nanopore sensor recorded at +60 mV. The typical Cys-Cu events are marked with “red triangle”. (d-e) Scatter plot and histogram of Cys-Cu events. The electrical recording is carried out in 1 M KCl buffer at pH 7.4.

Figure R2. Difference between LFn and horse ferritin in amino acid sequence from position 114 to 131. (a) Table of amino acid sequences of horse ferritin (PDB 2w0o) and LFn (PDB 2ffx). The colorful amino acids are different between horse ferritin and LFn. (b) Structure of C3 channel on horse ferritin (PDB:2w0o). The Gln120, Ala121, and Ser131 residues are highlighted. (c) The molecular structure difference between the above three amino acids.

Finally, to explore the effect of the iron core on the sensing property of ferritin, we added additional nanopore experiments using Sigma-Aldrich horse spleen apo-ferritin (iron-unloaded) which was characterized by HRTEM experiments (Supplementary Fig. 10a). As a result, the apo-ferritin can also be inserted into lipid bilayer and display the almost identical I-V characteristic as iron-loaded ferritin (Supplementary Fig. 10b). In addition, the apo-ferritin nanopore can also be used for Cys-Cu complex sensing (Supplementary Fig. 10c). The scatter plot and histogram of Cys-Cu events sensing with apo-ferritin have no obvious difference with that in iron-loaded ferritin testing (Supplementary Fig. 10d, e, Fig. R3). Therefore, we conclude that the sensing property for Cys-Cu is an intrinsic feature of ferritin protein shell and is independent of the internal iron core. The following sentence was added in the main text. “At the same time, compared with ferritin, the apo-ferritin without internal iron core exhibits the identical sensing behavior for Cys-Cu detection,

revealing that the conductive and sensing properties of ferritin nanopore are independent of the iron core loaded in ferritin, but attributed to the multi-channel structure and the suitable sensitive sites in the channels (Supplementary Fig. 10).” (Please see Page 5 left column, 1st paragraph with yellow highlight).

Supplementary Figure 10. Nanopore experiments with horse spleen apo-ferritin. (a) Stained HTREM images for ferritin (left) and apo-ferritin (right). Iron-loaded ferritin results in a darker central region, whereas apo-ferritin without iron core does not. (b) Comparison of the I-V curves of ferritin (black) and apo-ferritin (red), in which, almost identical I-V curves are observed. (c) Nanopore experiments for Cys-Cu (80 μ M) detection using an apo-ferritin sensor at +40 mV. (d-e) Scatter plot and histogram of Cys-Cu events. The signals are almost the same as that observed in using ferritin sensor. The nanopore recording is carried out in 1 M KCl buffer at pH 7.4.

Figure R3. Comparison of Cys-Cu events sensed by holo-ferritin nanopore and apo-ferritin. (a-b) Scatter plot and histogram of Cys-Cu events sensed by holo-ferritin. (c-d) Scatter plot and histogram of Cys-Cu events sensed by apo-ferritin. There are no obvious differences in nanopore signals between two batches of horse spleen ferritin. The nanopore recordings are carried out in 1 M KCl buffer at pH 7.4 under +40 mV.

2. The authors have partially addressed the concern regarding the multi-level signals and the relationship between analyte concentration and simultaneous pore blockages (Question #2). They provided data showing an increase in the proportion of multi-level blockage signals with higher concentrations of L-Cys (Supplementary Fig. 9). However, the original concern is about the calculation and analysis of multi-level events (i.e., simultaneous pore blockages) and how these events vary with analyte concentration. The Reviewer was interested in a quantitative approach to understand and predict the likelihood of multi-level signals. This addition does not address the statistical analysis or propensity of simultaneous blockages. The added experiment (Supplementary Fig. 10) does not address the frequency or behavior of multi-level signals explicitly, which was the core of the question. Additionally, the new data does not provide insights into the mechanistic basis of multi-level blockages or explain the multi-channel behavior of the ferritin nanopore.

Response 2: Thank you so much for your valuable comments. In this revision, we have built the quantitative mathematic approach to simulate and predict the multi-level signals by fitting the experimental data with the models.

To simulate the multi-channels of ferritin, we simplify it as four identical channels. For the four-channel ferritin nanopore, each of four channels has the same each-channel-blocked probability (q) and each-channel-opening probability ($1-q$). Theoretically, there are five possible states ($X=k, k=0,1,2,3,4$), where $P(X=k)$ represents the probability that k of the four channels is/are simultaneously blocked (Supplementary Fig. 9a). For example, $X=0$ means none channel is blocked, corresponding to current baseline (state 0), and $X=2$ indicates that two of the four channels are simultaneously blocked, which corresponds to the state 2 signals observed in nanopore recordings.

Frist, the experimental data about multi-level signals was summarized and analyzed. The experimental probabilities of five states ($P_{\text{exp}}(X=0,1,2,3,4)$) at five different Cys-Cu concentrations (0, 20, 40, 60, and 80 μM) were calculated from the experimental current traces. The experimental probability is defined as the dwell time proportion and summarized in Supplementary Table S1. Such as for state 1 signals at 80 μM , the experimental probability ($P_{\text{exp}}(X=1)$) is 0.17312, which is calculated by dividing the total dwell time of state 1 signals by the total recoding time. To be noted that, during the experiments under current Cys-Cu concentrations, no state 4 signals were observed and $P_{\text{exp}}(X=4)$ thus was zero.

Second, a binomial distribution model was established to fit the experimental data. For the

four-channel ferritin nanopore, each of four channels has the same each-channel-blocked probability (q) and each-channel-opening probability ($1-q$). The binomial distribution model $\text{Bin}(4,q)$ is suitable to describe the relationship between the occurrence probability $P(X=k)$ of multi-level signals and the q values. The binomial distribution model formula $\text{Bin}(4,q)$ is described as:

$$P(X = k) = \binom{4}{k} q^k (1 - q)^{4-k}, k = 0,1,2,3,4$$

$$\sum_{k=0}^4 \binom{4}{k} q^k (1 - q)^{4-k} = 1$$

in which, k means the number of simultaneously blocked channels, q means the each-channel-blocked probability of each channel, $P(X=k)$ means the probability that k of the four channels is/are simultaneously blocked, and the sum of all the probabilities of five states ($X=0,1,2,3,4$) equals to 1. According to above formula, the model curves depicting how the predicted probability of five states ($P_{\text{pre}}(X=0,1,2,3,4)$) changes as different q value were generated through python code (please see code files) (Supplementary Fig. 9b, c).

Third, for each Cys-Cu concentration, the experimental probabilities $P_{\text{exp}}(X=0, 1, 2, 3, 4)$ were respectively substituted into the model curves to fit an optimal q value. In this way, we obtained five q values ($q_0=0$, $q_{20}=0.0066$, $q_{40}=0.0198$, $q_{60}=0.0351$, and $q_{80}=0.0494$) corresponding to the Cys-Cu concentrations of 0, 20, 40, 60, and 80 μM , respectively (Supplementary Fig. 9c). As shown in Supplementary Fig. 9d, the data points corresponded quite well with the model curves, revealing the good credibility of obtained q values.

Finally, the data points of q values at different concentrations were plotted in Supplementary Fig. 9d. Then, the logistic model, usually used in describing the relationship between receptor response and ligand concentrations, was used to fit the relationship between each-channel-blocked probabilities (q values) and Cys-Cu concentrations. The logistic model was found to have a good performance in fitting their relationship. Therefore, the logistic model can be used to predict q value at a given analyte concentration. Then, according to the binomial distribution model curves, one can calculate the occurrence probabilities of each kind of multi-level signals.

The following sentence was added in the main text. “The multi-level signal of L-Cys, as an example, is analyzed, in which the quantitative relationship between the probability of multi-level signals and the Cys-Cu concentrations is described using binomial distribution model and logistic model (Supplementary Fig. 9b-d).” (Please see Page 5 left column, 1st paragraph with yellow

highlight).

For your information, as for the Supplementary Fig. 10, it is the response to Reviewer 1# (Question 9 in the first round) to display the qualification capability of ferritin nanopore for the actual sample (L-cysteine capsule), which is independent of the multi-level signal quantitative approach.

Supplementary Table S1. The experimental probabilities of five states (X=0,1,2,3,4) at five different Cys-Cu concentrations (0-80 μM)^a.

$P_{\text{exp}}(X=k)$ $C_{(\text{Cys-Cu})}$	$P_{\text{exp}}(X=0)$	$P_{\text{exp}}(X=1)$	$P_{\text{exp}}(X=2)$	$P_{\text{exp}}(X=3)$	$P_{\text{exp}}(X=4)$
80 μM	0.81867	0.17312	0.00706	0.00118	0
60 μM	0.86561	0.12520	0.00247	0.00324	0
40 μM	0.92366	0.07495	0.00120	0.00042	0
20 μM	0.97308	0.02534	0.00044	0.00006	0
0 μM	1	0	0	0	0

^aThe experimental probabilities for each concentration were defined as their respective dwell time proportions in nanopore recordings. At the same concentration, the sum of the probabilities of all states equals 1. For 0 μM Cys-Cu, no other signals but only baseline was observed. No current blockages of state 4 were observed.

Supplementary Figure 9. Models for multi-level signals analysis. (a) Schematic diagrams and current traces of multi-level signals. After a ferritin is inserted into the lipid bilayer, there are four C3 channels on a side of the lipid membrane. Current baseline means none channel is blocked. Single-level signal (state 1) means one of four channel is blocked. Multi-level signals (state 2, state 3, and state 4) mean multiple channels are simultaneously blocked. (b) Predicted probability curves of five states by the Bin(4, q) binomial distribution model. (c) Magnified view of (b) ranging from 0 to 0.1 on the x-axis. The lines are predicted model curves of the five states ($X=0, 1, 2, 3, 4$). The data points are actual experimental probabilities of the five states ($X=0, 1, 2, 3, 4$) in different Cys-Cu concentrations (0 to 80 μ M). (d) The relationship between the channel-blocked probability (q) and the Cys-Cu concentration, which is fitted to the logistic model.

3. Once again, the authors have sidestepped the original question and opted for an alternative approach that raises additional concerns. First, the authors cite a lack of time and expertise in bioengineering as reasons for not preparing an L-mutant ferritin. However, there are numerous bioengineering companies and suppliers capable of providing mutant ferritin within a relatively short timeframe. Second, the alternative approach that the authors introduced employed a large excess of zinc (0.24 mM), but it remains unclear where the added zinc binds—whether in the C3 channels or on the ferritin surface. The authors have assumed, without supporting evidence, that the zinc ions would replace the copper ions in the Cys-Cu complexes within the C3 channels. Thus, this experiment failed to address the reviewer's concern and to conclusively demonstrate the critical role of the three amino acids (His114, Cys126, and Glu130) in stabilizing the Cys-Cu complexes.

Response 3: Thanks for your valuable comments. In the first-round manuscript, we resulted that the nanopore signals are caused by Cys-Cu complex binding with His114, Cys126, and Glu130 residues in C3 channel through MD simulation (Fig. 4c, d). For this part, the reviewer #3 suggested us to shield these three residues to experimentally verify their key role in binding with Cys-Cu complex. In our first-round revision, we have tried our best to acquire the mutant ferritin from collaborators including the reviewer #3, however, we did not receive the mutant ferritin from them until the revision deadline. Mutant ferritin is a reasonable suggestion but not the only solution to shield these three residues. In the first-round revision, the excess Zn^{2+} was used as the shield agent to occupy these binding sites (His114, Cys126, and Glu130) to experimentally verify their key role in binding with Cys-Cu complex. We have to correct the reviewer that it is not our assumption that " Zn^{2+} has the same binding sites (His114, Cys126, and Glu130) as Cu^{2+} in the C3 channel", but rather a finding reported in previous studies, supported by clear crystal structure evidence (*PNAS*, 2014, **111**, 7925-7930) (Fig. R4).

In this revision, the further MD simulation was added to verify the stable binding between Zn^{2+} and these three residues (His114, Cys126, and Glu130). In MD simulation, the coordination of Zn^{2+} with the three residues His114, Cys126, and Glu130 is consistent with that of Cu^{2+} (please see Supplementary Fig. 17 for the MD simulation of Cu^{2+}), and the small RMSD value suggests that Zn^{2+} can stably reside at this position (Supplementary Fig. 19a, b). Furthermore, a concentration control experiment for Zn^{2+} was also conducted, and it was found that as the Zn^{2+} concentration increased, the frequency of Cys-Cu events dramatically decreased (Supplementary Fig. 19c). This

clearly indicates the competitive occupation effect of Zn^{2+} with Cys-Cu complex on the binding sites (His114, Cys126, and Glu130) in C3 channel.

As for the mutant protein without these three residues, we have contacted three biotechnology companies and our collaborator who support the LFn and HFn samples. However, we have not yet received it neither from our collaborators nor from the biotechnology companies until Round 2 deadline with additional twice one-month extensions. We consider it is unnecessary to wait further longer for the preparation of the time- and cost-consuming mutant ferritin as a negative control experiment, especially when we have verified the key role of these residues by combining MD simulations and Zn^{2+} concentration control experiments.

The following sentence was added in the main text. “To further verify the key roles of these residues, we introduce excess Zn^{2+} , that also has stronger interaction with these residues,³⁴ in the detection for Cys-Cu complex to occupy these binding sites and analyze how the Cys-Cu events frequency becomes before and after introducing Zn^{2+} . The MD simulation was first conducted to further confirm the coordination style of Zn^{2+} in C3 channel, in which, the Zn^{2+} was observed to indeed bind with these three residues and keep stable (Supplementary Fig. 19a, b). Furthermore, the concentration control experiments for Zn^{2+} were conducted, and it was found that the frequency of Cys-Cu events gradually decreased as the Zn^{2+} concentration increased, accordingly revealing the key roles of these residues (His114, Cys126, and Glu130) in binding with Cys-Cu complex (Supplementary Fig. 19c).” (Please see Page 7 right column, 1st paragraph with yellow highlight).

Fig. 4 Simulation of binding conformations of Cys-Cu-ferritin based on the multiscale (QM/AA/CG) model. **a**, A schematic diagram of the CG division map for different regions. Cys-Cu-Ferritin (Prot) is represented by three scales, QM, AA, and CG. Water solvents (Wat) are represented by two scales, AA and CG. Position restrictions (PR) are adopted for molecules near interfaces to prevent diffusions. **b**, The initial conformation of the active and non-active region in the Cys-Cu-Ferritin. The active region corresponds to the AA & QM regions in (a), represented by the cyan cartoon (protein) and AA water molecules. Non-active CG regions are depicted using white surfaces (protein) and green dots (solvents), respectively. **c**, The initial conformation of the QM region represented by Cu^{2+} , L-Cys, His114, Cys126, and Glu130 and two nearby water molecules. **d**, The root mean square deviation (RMSD) of Cu^{2+} and five coordinated atoms from residues with respect to the simulation time. Primary coordination patterns of corresponding RMSD are given above, with hydrogen bonds represented by dotted lines.

Fig. S2. Divalent metal ion inhibition of ferritin catalysis and binding sites in ferritin threefold ion channels. (A) Changes in the initial rates of DFP formation in WT ferritin (Table S5). (B) Divalent metal binding in the ion channels of ferritin protein cages. The channels are formed by residues from protein subunit helix 4 and H114 in helix 3. The position of Zn²⁺ ions and the conformation of C126 residues in the ferritin ion channels are identical to Cu²⁺ in ferritin protein structures. (Left) PDB file 3KA3 (Mg²⁺), (Center) PDB 3KA4 (Co²⁺), and (Right) PDB 3RE7 (Cu²⁺), using PYMOL. Spheres: green, Mg²⁺; pink, Co²⁺; reddish-brown, Cu²⁺.

Figure R4. Crystal structure displaying the binding between Cu²⁺ and three residues (His114, Cys126, and Glu130) reported by Behera *et al.*, [8] in which, “The position of Zn²⁺ ions and the conformation of C126 residues in the ferritin ion channels are identical to Cu²⁺ in ferritin protein structures.” was described.

[8] Behera, R.K. & Theil, E.C. Moving Fe²⁺ from Ferritin Ion Channels to Catalytic OH Centers Depends on Conserved Protein Cage Carboxylates. *Proc. Natl. Acad. Sci. U.S.A.* **111**, 7925-7930 (2014).

Supplementary Figure 17. MD simulation of the Cu^{2+} coordination style of Cu-protein complexes in C3 channel. (a) The reported crystal structure of Cu^{2+} -bonded ferritin (PDB 3RE7). To clearly show the coordination of Cu^{2+} , only one Cu^{2+} is retained. (b) The equilibrated ferritin- Cu^{2+} model showing the combination state of Cu^{2+} in the C3 channel. The single Cu^{2+} is observed to coordinate with His114, Cys126, and Glu130 in the C3 channel, which closely resembles the crystal structure of Cu^{2+} -bonded ferritin. (c) The RMSD plot showing relaxation process of the ferritin- Cu^{2+} model in MD simulation. The small RMSD value reflects the favorable stability of the Cu^{2+} coordination structure. The details of MD simulation have been described in the Methods section.

Supplementary Figure 19. Occupation effect of Zn²⁺ to the Cys-Cu binding sites. (a) MD simulation of the Zn²⁺ coordination style in ferritin C3 channel. The single Zn²⁺ is observed to coordinate with His114, Cys126, and Glu130, which is identical to the coordination style of Cu²⁺. (b) RMSD plot showing relaxation process of the ferritin-Zn²⁺ model. The small RMSD value reflects the favorable stability of the Zn²⁺ coordination structure. (c) Events frequency in Cys-Cu complex detection with Zn²⁺ in different concentrations. The frequency of Cys-Cu events decreased as the Zn²⁺ concentration increased, revealing the obvious occupation effect of Zn²⁺ to the Cys-Cu binding sites (His114, Cys126, and Glu130). All the nanopore experiments are carried out in 1 M KCl buffer with 80 μM Cu²⁺, 80 μM Cys and Zn²⁺ (0, 160, 240, or 320 μM) in trans chamber at +40 mV.

This completes our reply. We thank the Referees for their valuable comments/suggestions and critical reading of our work. We hope that the changes we made will prove satisfactory. At the same time, we remain open to further suggestions as to how to improve the manuscript.

With warm regards,

Zhi-Yuan Gu & Co-Authors

Response to Reviewer' Comments

Reviewer: 4

Comments: 1. The authors have thoroughly and satisfactorily addressed my questions and concerns, providing additional data and detailed explanations that have strengthened the study and made the manuscript's conclusions more robust. Although both HF_n and LF_n share the key coordinating residues in the C3 channel (His114, Cys126, and Glu130), differences in their overall structure and electrostatic or surface properties appear to have prevented HF_n from embedding in the membrane and functioning as a nanopore under the tested conditions. This is likely due to variations in surface charge and hydrophobicity, which are critical for membrane insertion and nanopore activity. I recommend that the authors include this information in the main manuscript for clarity. This result represents a novel and important finding that emerged only after the second round of reviews.

Response 1: Thank you so much for your time, effort, and expertise in reviewing our work. Your constructive suggestions greatly improved the quality of our manuscript. As you suggested, we have added relevant description in the main manuscript. The following sentence was added in the main text: It is worth noting that the L-subunit of ferritin appears to play a key role in facilitating the embedding of ferritin into lipid bilayers. In our experiments, both L-rich horse ferritin and recombinantly expressed human L-ferritin (LF_n) could successfully embed into lipid bilayers, whereas recombinantly expressed human H-ferritin (HF_n) consistently failed to incorporate despite multiple attempts. (Please see **Page 3 right column, 1st paragraph with yellow highlight**)

2. I would also encourage the authors to consider investigating a range of H/L subunit compositions (in a future study) to further optimize sensor performance. A blend of H and L subunits could potentially provide improved properties, such as enhanced membrane embedding, stability, or selectivity, compared to L-rich ferritin alone, offering valuable insights into how subunit composition affects nanopore behavior and analyte recognition. This is particularly interesting since physiological ferritin is typically a heteropolymer with subunit composition relevant to numerous disease states (e.g. iron metabolism disorders, cancer, and neurodegeneration). Such research is feasible given the recent advances in synthesizing and purifying a range of H/L subunit compositions and could help bridge fundamental nanopore science with clinical applications, making these findings more broadly impactful and translational.

Response 2: Thanks for your suggestions. You guide us an interesting research topic and we will further optimize the ferritin sensor performance by regulating the ratio of H/L subunits in the future.

This completes our reply. We thank the Referees for their valuable comments/suggestions and critical reading of our work. We hope that the changes we made will prove satisfactory. At the same time, we remain open to further suggestions as to how to improve the manuscript.

With warm regards,

Zhi-Yuan Gu & Co-Authors